# Understanding Graph Transformers by Generalized Propagation

## Abstract

Graph Transformers (GTs) have recently shown stellar performance on various graph learning benchmarks, which is typically attributed to their underlying global self-attention mechanism. In this paper, we use generalized propagation graphs, constructed through two abstract configurable functions and offering a unified view across various GNN models used in the literature. We show that by configuring the two abstract functions governing the generation of propagation graph, one could recover the most popular GNN models including graph gransformers, message-passing neural networks (MPNNs), as well as various forms of graph rewiring. We show that the expressivity of the instances of our framework depends on one of the governing functions (the adjacency function). Empirical results confirm our theory: by keeping the adjacency function while removing self-attention, the state-of-the-art GT maintains its performance. In other words, by designing appropriate adjacency functions, one could construct novel GNN models with diverse expressive power. We also study the geometric properties of the propagation graphs across a wide range of models, using a novel extension of the Ollivier-Ricci curvature to weighted digraphs.

## 1 Introduction

Graph-structured data is ubiquitous across diverse domains of science, such as biomedicine (Zitnik et al., 2018), physics (Shlomi et al., 2021), chemistry (Duvenaud et al., 2015), material science, chip design, and social networks. Analyzing such datasets with data-driven methods has been the focus of geometric deep learning (Bronstein et al., 2021). Pioneering studies in graph neural networks (GNNs) aimed to generalize recurrent (Gori et al., 2005; Scarselli et al., 2009) and convolutional (Bruna et al., 2014; Defferrard et al., 2016; Kipf & Welling, 2017) architectures to graphs. These methods were later shown to be special instances of the message-passing architecture (Gilmer et al., 2017), which acts by propagating information between adjacency nodes of the input graph. Today, message-passing neural networks (MPNNs) and their variants (Hamilton et al., 2017; Veličković et al., 2018; Monti et al., 2017) are arguably the most common GNN architecture (Veličković, 2022) and have reached outstanding performance in various graph learning tasks.

However, MPNNs are known to suffer from over-squashing (Alon & Yahav, 2020) and over-smoothing (Chen et al., 2020; Oono & Suzuki, 2019) phenomena. Furthermore, the expressive power of message passing is bounded by the first-order Weisfeiler-Lehman test (1-WL) (Xu et al., 2019; Chen et al., 2019), which makes it impossible to discriminate between certain non-isomorphic graph pairs. Various methods have been proposed to overcome those limitations. *Graph rewiring* approaches (Topping et al., 2022; Gutteridge et al., 2023) decouple the input graph from the computational one, in order to make the graph 'friendlier' for message passing and alleviate over-squashing phenomena. *Higher-order GNNs* follow more expressive $k$-WL tests (Maron et al., 2019; Morris et al., 2019b). *Subgraph GNNs* (Bevilacqua et al., 2022; Alsentzer et al., 2020; Frasca et al., 2022) operate on a collection of subgraphs extracted by some policy, in order to achieve higher expressivity.

On the other hand, Transformers (Vaswani et al., 2017b), which have become predominant in natural language processing and computer vision, can also be seen in principle as MPNNs on a complete graph that is learned through the attention mechanism. However, since Transformers do not directly work well for graphs (Dwivedi & Bresson, 2021), recent works have studied various Graph Trans-

former (GT) architectures, from integrating with MPNNs (Kreuzer et al., 2021b; Rampášek et al., 2022; Chen et al., 2022; Gutteridge et al., 2023) to introducing various positional encoding (Ying et al., 2021; Zhang et al., 2023b; Ma et al., 2023; Kim et al., 2022). While these efforts have yielded superior performance on various benchmarks and diverse heuristics, we still lack an understanding of how graph Transformers work and when they are advantageous over MPNNs.

**Main contributions.**  In this paper, we introduce a unified view by giving a definition to the *generalized propagation graph*, a weighted directed graph constructed from the input graph but not necessarily identical to it. More specifically, propagation construction is governed by two functions – *adjacency function* $f(\boldsymbol{A})$ and *entry-wise function* $\pi(\mathbf{X}_u, \mathbf{X}_v, f(\boldsymbol{A}))$, where $\boldsymbol{A}$ is the adjacency matrix of the input graph and $\mathbf{X}_u, \mathbf{X}_v$ denote the features of nodes $u$ and $v$. By configuring $f$ and $\pi$, we unify the various GNN families, from MPNNs, graph rewiring, and SubgraphGNNs to GTs, into a general framework, *generalized propagation neural networks* (GPNNs). Rooting from GPNN, we develop two theoretical contributions.

We show that the expressiveness of models within GPNN framework sorely depends on $f(\boldsymbol{A})$. This result could be used in two ways: (i) by designing adjacency functions, one could explore novel graph models with diverse expressive power; and (ii) by inspecting $f(\boldsymbol{A})$ of an existing model, one can decide its expressiveness upper-bound.

Second, we show that GPNN facilitates comparative analysis of their strengths and weaknesses by studying the geometric properties (in particular, discrete curvature that has previously been used for graph rewiring (Topping et al., 2022)) of the propagation graphs and how information flows on them. For this purpose, we extend the Ollivier-Ricci (OR) curvature (Ollivier, 2009) to weighted directed graphs. To facilitate direct comparison across models and graphs, our design enjoys two benign analytical properties: continuity and scale-free. Thus we name our extended OR curvature as *Continuous Unified Ricci Curvature* (CURC).

Third, to test that expressiveness solely depends on $f(\boldsymbol{A})$, we introduce a variant of GPNN we call GPNN-PE by keeping the adjacency function while removing self-attention (as a special design of $\pi$) from one of the state-of-the-art GTs, GRIT (Ma et al., 2023). We show that on a wide range of benchmarks, GPNN-PE matches the performance of GRIT with fewer parameters and less computation, being consistent with our theory. It is worth noting that our empirical findings question the belief that graph transformers benefit from global self-attention (Vaswani et al., 2017a; Park & Kim, 2021).

Finally, utilizing CURC, we visualize the curvature distribution of learned propagation and compare it with MPNN propagation, revealing the information flow of the underlying model. Such quantitative investigation across a wide range of GTs - to the best of our knowledge - has not been studied in the literature.

## 2  PRELIMINARY AND BACKGROUND

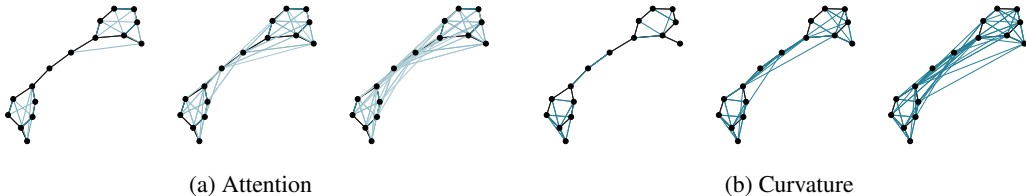

(a) Attention                              (b) Curvature

Figure 1: The Demonstration of Top-$10/20/30\%$ Attention and CURC-curvature on a ZINC graph (Irwin et al., 2012). The attention is obtained with the **GPNN-PE(1-head)** model.

Let $\mathcal{G} = (\mathcal{V}, \mathcal{E})$ be a graph with nodes $\mathcal{V} = \{1, \ldots, n\}$ and edges $\mathcal{E} \subseteq \mathcal{V} \times \mathcal{V}$. The structure of the graph is represented by the $n \times n$ *adjacency matrix* $\mathbf{A}$ where $a_{uv} = 1$ iff $(u, v) \in \mathcal{E}$ and zero otherwise. We further assume the graph to be attributed and represent by $\mathbf{X} \in \mathbb{R}^{n \times d}$ and $\mathbf{E} \in \mathbb{R}^{n^2 \times d}$ the node and edge attributes, respectively, which we assume w.l.o.g. to be $d$-dimensional. Unless specified, the dimension $d$ is excluded from matrix multiplication or tensor contraction. Any

operations involving dimension $d$ will be explicitly denoted with a superscript, e.g., $\mathbf{W}^d$ represents a linear operator solely on the $d$-dimension, broadcast across other dimensions.

## 3 GENERALIZED PROPAGATION FRAMEWORK

### 3.1 GENERALIZED PROPAGATION GRAPH

We start with the definition of propagation graph [1] which lies in the center of our framework. A propagation graph is a generalized abstraction of both the attention in graph transformers as well as message passing graphs in MPNNs.

**Definition 3.1** (Propagation Graph). A propagation graph is an edge-weighted directed graph (digraph) defined by a triplet $\mathcal{P} = (\mathcal{V}, \mathcal{E}, \omega)$ consisting of the vertex set $\mathcal{V}$, the set of directed edges with self-loops $\mathcal{E} \subseteq \mathcal{V}^2$ and the weight function $\omega : \mathcal{V}^2 \mapsto \mathbb{R}^+$.

In this work, we do not discriminate between $(u, v) \notin \mathcal{E}$ and $\omega(u, v) = 0$. Thus one can always assume that $\mathcal{E} = \mathcal{V}^2$ and leave the definition of the propagation graph to the weight function. The definition is valid for both undirected graphs and unweighted graphs since they could be regarded as subsets of weighted digraphs. Undirected graphs impose the restriction of **symmetric weight**, $\omega(u, v) = \omega(u, v)$. Unweighted graphs impose the restriction of **binary weight**, $\omega : \mathcal{V}^1 \mapsto \{0, 1\}$. Note that the propagation graph is uniquely defined by its weight function and space of weight functions $\mathcal{W} = \mathrm{Hom}(\mathcal{V}^2, \mathbb{R}^+)$ being isomorphic to $\mathbb{R}^{+^{n^2}}$. Thus the propagation is uniquely defined by the matrix $\boldsymbol{P} \in \mathbb{R}^{+^{n^2}}$.

### 3.2 GENERALIZED PROPAGATION NEURAL NETWORK (GPNN)

Here, we introduce the adjacency function and entries-wise function governing the generation of propagation graphs. The definition will depend on the permutation equivariant mapping over some tensor power space $\mathbb{R}^{n^k}$. We refer readers to Appendix A.1 for a detailed introduction. First, we define *adjacency function*, denoted as $f$, as a permutation equivariant mapping from $\boldsymbol{A}$ to $\mathbb{R}^{n^2 \times d}$. Then, we define *entry-wise function*, denoted as $\pi$, as a mapping from $\mathbb{R}^d$ to $\mathbb{R}$.

Other parameterized functions in our framework are $\phi^d : \mathbb{R}^d \mapsto \mathbb{R}^d$ only on feature dimension $d$. By applying $\phi^d$ to each feature in $\boldsymbol{X}$, we define $\phi : \mathbb{R}^{n \times d} \mapsto \mathbb{R}^{n \times d}$. And $\rho$, which represents the normalized function. And an update function $U : R^{n \times d} \times R^{n \times d} \mapsto R^{n \times d}$, applies node-wise update similar to MPNN.

Now we define the GPNN layer which takes the adjacency and node feature $\boldsymbol{A}$ as the input $\boldsymbol{X}$ and produces the updated node features $\boldsymbol{X}'$. Here, we show the single layer and single head version:

$$\boldsymbol{X}' = U(\rho(\boldsymbol{P}) \times \phi(\boldsymbol{X}), \boldsymbol{X})$$
$$\text{where } \boldsymbol{P}_{uv} = \pi(\boldsymbol{X}_u, \boldsymbol{X}_v, f(\boldsymbol{A})_{u,v}) \tag{1}$$

where $\times$ denotes the matrix multiplication between normalized propagation $\rho(\boldsymbol{P})$ and $\phi(\boldsymbol{X})$. Just like self-attention, we can also define the multi-head GPNN as

$$\boldsymbol{X}^{l+1} = U([\rho(\boldsymbol{P}^h) \times \phi^{h,l}(\boldsymbol{X})]_{\mathrm{concat}}^h, \boldsymbol{X})$$
$$\text{where } \boldsymbol{P}_{uv}^h = \pi^{h,l}(\boldsymbol{X}_u^l, \boldsymbol{X}_v^l, f^{h,l}(\boldsymbol{A})_{u,v}) \tag{2}$$

and $[\ ]_{\mathrm{concat}^h}$ denotes concatenating tensors indexed by $h$ along the feature dimension. In Table 1, we list three examples of casting existing methods into the GPNNs framework. More results can be fined in Appendix D.

### 3.3 EXPRESSIVENESS OF GPNNS

The expressiveness of GPNNs varies based on what exact functions are used. Thus a general expressiveness assessment based on abstract form would not be much of interest. However, to ease

---

[1] The *propagation* in our context follows the *forward-propagation* concept introduced in previous literature (Pearl, 1988; Minka, 2001; Yedidia et al., 2003; Hamilton et al., 2017) and differs from *back-propagation* algorithm (Rumelhart et al., 1986)

the expressiveness assessment for methods falling into the GPNN category, we derive two expressiveness results for two prototypical GPNN models. First, we consider the GPNNs with identical adjacency functions for different heads and layers. We refer to the resulting model as **Homogeneous GPNNs**. The layer update of **Homogeneous GPNNs** is identical to 2 except

$$f^{h,l} = f \tag{3}$$

For generality, we also consider the case when multiple adjacency function $f$ is used in the model. An extra prototype model is defined with recurrence as **Layer-recurrent GPNNs** by defining:

$$
\begin{aligned}
f^{h,l} &= f^l \\
f^{l+p} &= f^l
\end{aligned}
\tag{4}
$$

In order to reach the upper-bounded expressiveness, the model would be assumed with a sufficient number of heads and two **MLPs** with a sufficient layer and width for $\pi$ and $\phi$, and update function $U$.

**Proposition 3.1.** *Homogeneous GPNNs Suppose a Homogeneous GPNN model has a fixed propagation function $f(\boldsymbol{A})$, with sufficient heads and layers, the expressiveness of Homogeneous GPNN is upper-bounded by the iterative color-refinement*

$$\mathcal{X}_{\mathcal{G}}^{t+1}(v) = hash\{\{(\mathcal{X}_{\mathcal{G}}^t(u), f(A)_{vu}) : u \in \mathcal{V}\}\} \tag{5}$$

**Proposition 3.2.** *Layer-recurrent GPNNs Suppose a Layer-recurrent GPNN model M has a layer-dependent propagation function $f^l(\boldsymbol{A})$, repeats every $p$ layer: $f^l = f^{l+p}$. With sufficient heads and layers, by stacking repetitions of such repetition, the expressiveness of M is upper-bounded by the iterative color refinement*

$$\mathcal{X}_{\mathcal{G}}^{t+1}(v) = hash\{\{\left(\mathcal{X}_{\mathcal{G}}^t(u), \left(f^1(A)_{vu}, f^2(A)_{vu}, \ldots, f^p(A)_{vu}\right)\right) : u \in \mathcal{V}\}\} \tag{6}$$

For detailed proof, see Appendix C.2 and C.2. Here, we would like to highlight that Homogenous GPNNs contain interesting variants like MPNNs, Graphormer, or layer-independent graph rewiring, while Layer-recurrent GPNNs contain GraphGPS (Rampášek et al., 2022). By configuring $f(\boldsymbol{A})$, one can easily modify the upper-bounding color refinement iteration (see Appendix C.1 for more details). The WL test appears to be too coarse to be used to evaluate the expressive power of current graph models, covered by (Morris et al., 2022). Other than the color refinement algorithm, a new expressiveness hierarchy has been proposed (Puny et al., 2023; Zhang et al., 2023a; Zhou et al., 2023; Geerts & Reutter, 2022). Moreover, we would like to particularly emphasize the connection between equivariant polynomials with the propagation function $f(\boldsymbol{A})$. Since $f$ is any permutation equivariant function in $\mathbb{R}^{n^2} \mapsto \mathbb{R}^{n^2}$, it can be approximated uniformly by the equivariant polynomial.

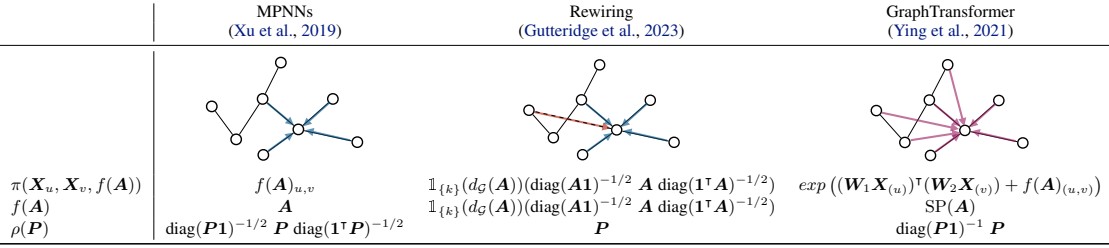

| | MPNNs (Xu et al., 2019) | Rewiring (Gutteridge et al., 2023) | GraphTransformer (Ying et al., 2021) |
|---|---|---|---|
| $\pi(\boldsymbol{X}_u, \boldsymbol{X}_v, f(\boldsymbol{A}))$ | $f(\boldsymbol{A})_{u,v}$ | $\mathbb{1}_{\{k\}}(d_{\mathcal{G}}(\boldsymbol{A}))(\mathrm{diag}(\boldsymbol{A1})^{-1/2}\,\boldsymbol{A}\,\mathrm{diag}(\boldsymbol{1}^\intercal\boldsymbol{A})^{-1/2})$ | $exp\left((\boldsymbol{W}_1\boldsymbol{X}_{(u)})^\intercal(\boldsymbol{W}_2\boldsymbol{X}_{(v)}) + f(\boldsymbol{A})_{(u,v)}\right)$ |
| $f(\boldsymbol{A})$ | $\boldsymbol{A}$ | $\mathbb{1}_{\{k\}}(d_{\mathcal{G}}(\boldsymbol{A}))(\mathrm{diag}(\boldsymbol{A1})^{-1/2}\,\boldsymbol{A}\,\mathrm{diag}(\boldsymbol{1}^\intercal\boldsymbol{A})^{-1/2})$ | $\mathrm{SP}(\boldsymbol{A})$ |
| $\rho(\boldsymbol{P})$ | $\mathrm{diag}(\boldsymbol{P1})^{-1/2}\,\boldsymbol{P}\,\mathrm{diag}(\boldsymbol{1}^\intercal\boldsymbol{P})^{-1/2}$ | $\boldsymbol{P}$ | $\mathrm{diag}(\boldsymbol{P1})^{-1}\,\boldsymbol{P}$ |

Table 1: We pick three methods from existing literature coming from three different model families to showcase the generality of the GPNN framework. We show that by defining different adjacency function $f$ and entry-wise function $\pi$ (with the help of $\rho$ to rescale) we can recover GIN, DREW, and Graphormer. $\mathrm{SP}(\boldsymbol{A})$ denotes the shortest path distance of graph.

## 4 CONTINUOUS UNIFIED RICCI CURVATURE

To analyze the over-squashing and bottlenecks in MPNNs, prior work (Topping et al., 2022) introduces the Balanced Forman curvature for measuring information flow in observed graphs. However, such curvature is limited to unweighted-undirected graphs, hindering our understanding of weighted-directed propagation graphs in the GPNN framework. Hence, we propose a continuous extension of the Ollivier-Ricci curvature applicable to any strongly-connected weighted-directed graphs.

## 4.1 CONNECTION BETWEEN CURC AND MESSAGE-PASSING

We identify the inherent relationship between information flow in a message-passing framework on graphs $G = (\mathcal{V}, \mathcal{E})$ and curvatures depending on optimal transportation like Ollivier-Ricci (OR) curvature. For nodes $x, y$, OR-curvature, represented as $\kappa_{OR}(x, y)$, evaluates the ratio of Wasserstein to graph distance, signifying the cost of moving uniform mass across edges. In this framework, information from each node $x$ diminishes as it spreads to neighboring nodes due to the nested aggregation function. Such diminishing impact is akin to the cost in Wasserstein distance. By normalizing this, we assert OR-curvature as a metric to measure the difficulty of information flow between nodes. This approach, unique compared to Forman-Ricci and resistance curvature, motivates us to develop a tool to study information flow in strongly-connected weighted-directed graphs $\mathcal{G} = (\mathcal{V}, \mathcal{E}, \omega)$ within GPNN.

Information flow in GPNN inherently differs from its MPNN counterpart for the following reasons:

- The distribution of information emanating from each node is uneven.
- Traversing different edges causes different extent of diminishing effect on information flow.
- Weighted-directed graphs exhibit distinct geometric and spectral properties when compared to unweighted-undirected graphs.

To address these distinctions and accurately model information propagation on weighted-directed graphs, we introduce the *Continuous Unified Ricci Curvature* as an extension to the Ollivier-ricci curvature, specifically designed for strongly-connected weighted-directed graphs. In the context of graph curvature, we adopt the conventional notations of $x$ and $y$ for nodes, as opposed to $u$ and $v$.

## 4.2 CONSTRUCTION OF CURC

**Definition 4.1.** On a strongly-connected weighted-directed graph $\mathcal{G} = (\mathcal{V}, \mathcal{E}, \omega)$, where $|\mathcal{V}| = n$. We define the function $\omega : \mathcal{V} \times \mathcal{V} \mapsto \mathbb{R}^{\geq 0}$ as edge weights. Note that $\omega$ is not necessarily symmetric, and we let $\omega(x, y) = 0$ to indicate $x \not\rightarrow y$. We define weighted out-degree matrix $D := \text{diag}(\omega_i)_{i=1}^n$, where $\omega_i := \sum_{j=1}^n \omega_{ij}$ and $W := D^{-1}\omega$ to be the random walk matrix. According to the *Perron-Frobenius theorem*, there exists a strictly positive left eigenvector $\mathbf{v}_{pf} \in \mathbb{R}^n$ of $W$. We define the *Perron measure* $\mathfrak{m} : \mathcal{V} \mapsto (0, 1]$ by normalizing $\mathbf{v}_{pf}$:

$$\mathfrak{m} := \frac{\mathbf{v}_{pf}}{\|\mathbf{v}_{pf}\|}.$$

Hence, we define the *mean transition probability kernel* $\mu : \mathcal{V} \times \mathcal{V} \mapsto [0, 1]$ by

$$\mu_x(y) = \mu(x, y) := \begin{cases} \frac{1}{2}[W(x, y) + \frac{\mathfrak{m}(y)}{\mathfrak{m}(x)} W(y, x)] & \text{if } y \neq x \\ 0 & \text{if } y = x \end{cases} \tag{7}$$

where $\sum_{y \in \mathcal{V}} \mu(x, y) = \sum_{y \in \mathcal{V}} \mu_x(y) = 1$ for fixed $x \in \mathcal{V}$.

**Definition 4.2.** Let $\varepsilon$ be a small positive real number, then the $\varepsilon$-*masked weighted edge length* $\mathfrak{l}^\varepsilon : \mathcal{V} \times \mathcal{V} \mapsto \mathbb{R}^{>0} \cup \{\infty\}$ is defined by

$$\mathfrak{l}^\varepsilon(x, y) := \begin{cases} \frac{1}{\omega(x, y)} & \text{if } \omega(x, y) \geq \varepsilon \\ \frac{1}{\varepsilon} & \text{if } \omega(x, y) < \varepsilon \end{cases}$$

We define the corresponding $\varepsilon$-*masked distance function* $d^\varepsilon : \mathcal{V} \times \mathcal{V} \mapsto \mathbb{R}^{>0} \cup \{\infty\}$ as

$$d^\varepsilon := \text{shortest weighted path with } \mathfrak{l}^\varepsilon \text{ as edge length,}$$

where $d^\varepsilon$ is a possibly asymmetric distance function on $\mathcal{V}$.

**Definition 4.3.** Let $\mathcal{G} = (\mathcal{V}, \mathcal{E}, \omega)$ be a strongly-connected weighted-directed graph, equipped with the $\varepsilon$-*masked weighted distance function* $d^\varepsilon$. For distinct $x, y \in \mathcal{V}$, we define the $\varepsilon$-*masked Continuous Unified Ricci Curvature* by

$$\kappa_{\text{CURC}}^\varepsilon(x, y) := 1 - \frac{\mathcal{W}_1^\varepsilon(\mu_x, \mu_y)}{d^\varepsilon(x, y)},$$

where the Wasserstein distance $\mathcal{W}_1^\varepsilon$ is based on $d^\varepsilon$. Hence, the *Continuous Unified Ricci Curvature (CURC)* is defined as

$$\kappa_{\text{CURC}}(x,y) := \lim_{\varepsilon \to 0} \kappa_{\text{CURC}}^\varepsilon(x,y).$$

**CURC measuring GPNN** In the GPNN framework, information propagation depends on the magnitude of edge weights rather than uniform distribution as in MPNN. The concept of *mean transition probability* encapsulates this property, effectively accounting for both stable distribution and the influence of outward edge weights. Throughout intermediate propagation steps, employing the reciprocal of edge weights as edge distances (costs) reflects the phenomenon that information attenuates more rapidly when traversing edges with lower weights. Furthermore, the incorporation of the Perron measure as a stable probability distribution imbues CURC with a geometric interpretation related to bottlenecking within weighted-directed graphs

## 4.3 ALGEBRAIC AND GEOMETRIC PROPERTIES OF CURC

**Proposition 4.1.** *The Continuous Unified Ricci Curvature $\kappa_{\text{CURC}}$ admits the following properties:*

- *For connected unweighted-undirected graph $\mathcal{G} = (\mathcal{V}, \mathcal{E})$, for any node pair $x, y \in \mathcal{V}$, we have $\kappa_{\text{CURC}}(x,y) = \kappa_{OR}(x,y)$, where $\kappa_{OR}$ is the Ollivier-ricci curvature.*

- *If we perceive $\kappa_{\text{CURC}}(x,y)$ as a function of $\omega \in \mathbb{R}^{n \times n}$, then $\kappa_{\text{CURC}}(x,y)$ is continuous w.r.t. $\omega$ entry-wise.*

- *For strongly-connected weighted-directed graph $\mathcal{G} = (\mathcal{V}, \mathcal{E}, \omega)$, when all edge weights $\omega$ are scaled by an arbitrary positive constant $\lambda$, the value of $\kappa_{\text{CURC}}(x,y)$ for any node pair $x, y \in \mathcal{V}$ is invariant.*

**Implication of algebraic properties of CURC** The first and second properties collectively establish CURC as a continuous extension of the canonical Ollivier-Ricci curvature, as originally introduced in (Ollivier, 2009). The third property further signifies that within the GPNN framework, if we uniformly scale all information propagations, the relative information flow remains unchanged, as evidenced by the invariance of curvature values.

**Theorem 4.2.** *Let $\mathcal{G} = (\mathcal{V}, \mathcal{E}, \omega)$ be a strongly-connected weighted-directed graph with (asymmetric) distance function $d : \mathcal{V} \times \mathcal{V} \mapsto \mathbb{R}^{\geq 0}$ satisfying triangle inequality and admits $d(x,x) = 0$ for all $x \in \mathcal{V}$. Then for probability measure $\mu, \nu : \mathcal{V} \mapsto [0,1]$, the **Kantorovich duality** holds. Namely,*

$$\inf_{\pi \in \Pi(\mu,\nu)} \sum_{x,y \in V} d(x,y)\pi(x,y) = \sup_{f \in \text{Lip}_1(\mathcal{V})} \sum f(z)\,(\mu - \nu). \tag{8}$$

*$\pi \in \Pi(\mu,\nu)$ is a coupling between $\mu, \nu$ and $f : \mathcal{V} \mapsto \mathbb{R} \in \text{Lip}_1(\mathcal{V})$, if for all $x, y \in \mathcal{V}$, $f(y) - f(x) \leq d(x,y)$.*

**Implication of KR duality** We pinpoint this specific condition for the choice of distance measures on $\mathcal{G}$ that enables KR duality to take effect and this transformation shifts the optimal transportation problem into a linear programming problem. Additionally, we offer two lower-bound estimations for CURC under distinct assumptions in proposition B.17 and B.19, with computational costs of $\mathcal{O}(n^3)$ and $\mathcal{O}(n^4)$, respectively.

**Definition 4.4.** For a non-empty $\Omega \subset \mathcal{V}$, its boundary *Perron-measure* is defined as

$$\mathfrak{m}(\partial\Omega) := \sum_{y \in \Omega} \sum_{z \in V \setminus \Omega} \mathfrak{m}_{yz},$$

where $\mathfrak{m}_{yz} := \mathfrak{m}(y)\mu(y,z)$ and $\mathfrak{m}(\Omega) = \sum_{x \in \Omega} \mathfrak{m}(x)$. Then the *Dirichlet isoperimetric constant* $\mathcal{I}_\mathcal{V}^D$ on $\mathcal{V}$ is defined by

$$\mathcal{I}_\mathcal{V}^D := \inf_\Omega \frac{\mathfrak{m}(\partial\Omega)}{\mathfrak{m}(\Omega)}.$$

**Theorem 4.3.** *Let $\mathcal{G} = (\mathcal{V}, \mathcal{E}, \omega)$ be a strongly-connected weighted-directed graph and $E_R(x) := \{y \in \mathcal{V} \mid d(x,y) \geq R\}$. Fix $x \in \mathcal{V}$, we assume $\inf_{y \in \mathcal{V} \setminus \{x\}} \kappa_{\text{CURC}}(x,y) \geq K$ for some $K \in \mathbb{R}$*

*and* $-\sum_{y \in \mathcal{V}} \mu(x, y) f(y) \geq \Lambda$ *for some* $\Lambda \in (-\infty, 0)$. *For* $D > 0$, *we further assume that for all* $y \in \mathcal{V}$, $d(x, y) \leq D$. *Then the Dirichlet isoperimetric constant admits the following lower bound:*

$$\mathcal{I}_{E_R(x)}^D \geq \frac{KR + \Lambda}{D}.$$

**Geometric implication of CURC** The *Dirichlet isoperimetric constant*, calculated utilizing the Perron measure, acts as an extension of the Cheeger constant for weighted-directed graphs. This constant is instrumental in delineating community information and in measuring bottleneck phenomena within these graphs. Intuitively, we can control the magnitude of $\mathcal{I}_{\mathcal{V}}^D$ when global CURC is bounded below by a positive number. This observation implies that as the CURC value increases, the likelihood of significant bottlenecking or the division of communities diminishes.

## 5 EXPERIMENT

### 5.1 THEORY-GUIDED DESIGN

In sec. 3.3, we theoretically demonstrate that the expressive power on distinguishing graphs of GPNN solely depends on the adjacency function $f$. We would like to show how such intuition could be utilized to design novel models. We propose a simplified variant of the state-of-the-art graph transformers - GRIT, termed **GPNN-PE**, which drops the irrelevant components to the adjacency function in the generalized propagation architecture. This model belongs to the Homogeneous GPNN family with $f(\boldsymbol{A}) = RRWP(\boldsymbol{A})$ which is a relative positional encoding based on random walk (see Appendix E.1.2). According to Prop. 3.1, the removal of self-attention would not change the adjacency function and, thus would not change its expressiveness power. The empirical results are shown in Table 3 and Table 2, showcasing that even with fewer learnable parameters and computation, GPNN-PE reaches compara-

Table 2: Test performance on LRGB (Dwivedi et al., 2022b). Shown is the mean±s.d. of 4 runs. Highlighted are the top first, second, and third results. # Param $\sim 500K$ for both datasets.

| Model | Peptides-func | Peptides-struct |
|---|---|---|
| | **AP↑** | **MAE↓** |
| GCN | $0.5930 \pm 0.0023$ | $0.3496 \pm 0.0013$ |
| GINE | $0.5498 \pm 0.0079$ | $0.3547 \pm 0.0045$ |
| GatedGCN | $0.5864 \pm 0.0035$ | $0.3420 \pm 0.0013$ |
| GatedGCN+RWSE | $0.6069 \pm 0.0035$ | $0.3357 \pm 0.0006$ |
| Transf.+LapPE | $0.6326 \pm 0.0126$ | $0.2529 \pm 0.0016$ |
| SAN+LapPE | $0.6384 \pm 0.0121$ | $0.2683 \pm 0.0043$ |
| SAN+RWSE | $0.6439 \pm 0.0075$ | $0.2545 \pm 0.0012$ |
| GPS | $0.6535 \pm 0.0041$ | $0.2500 \pm 0.0012$ |
| GRIT | $0.6988 \pm 0.0082$ | $0.2460 \pm 0.0012$ |
| GPNN-PE | $0.6954 \pm 0.0023$ | $0.2474 \pm 0.0010$ |
| GPNN-PE(share $\boldsymbol{P}$) | $0.6955 \pm 0.0057$ | $0.2454 \pm 0.0003$ |
| GPNN-PE(1-head) | $0.6874 \pm 0.0161$ | $0.2473 \pm 0.0013$ |

ble performance with GRIT in most cases. Further details concerning the experimental setup and hyperparameters can be found in Appendix E.2.

### 5.2 ABLATION: THE EFFECT OF SHARING $\pi$

We change the $\pi$ function by making it share weights across layers or share across layer and heads, leading to two more variants: **GPNN-PE(share $P$)** and **GPNN-PE(1-head)**. By the results in Table 3 and 2, we further confirmed that the design of $\pi$ is less important. Models with shared $\pi$ remains competitive compared with baseline, with much less parameters and computation expense.

#### 5.2.1 EVALUATION

We evaluate models on five datasets from the Benchmarking GNNs work (Dwivedi et al., 2022a) and two datasets from the Long-Range Graph Benchmark (LRGB) (Dwivedi et al., 2022b). These datasets are among the most widely used graph benchmarks and cover diverse graph learning tasks, including node classification, graph classification, and graph regression, with a focus on graph structure and long-range dependencies. Further details concerning the experimental setup can be found in Appendix E.2.

Table 3: Test performance in five benchmarks from (Dwivedi et al., 2022a). Shown is the mean $\pm$ s.d. of 4 runs with different random seeds. Highlighted are the top first, second, and third results. # Param $\sim 500K$ for ZINC, PATTERN, CLUSTER and $\sim 100K$ for MNIST and CIFAR10. * indicates statistically significant difference against the second-best result from the two-sample one-tailed t-test.

| Model | ZINC | MNIST | CIFAR10 | PATTERN | CLUSTER |
|---|---|---|---|---|---|
| | MAE↓ | Accuracy↑ | Accuracy↑ | Accuracy↑ | Accuracy↑ |
| GCN | $0.367 \pm 0.011$ | $90.705 \pm 0.218$ | $55.710 \pm 0.381$ | $71.892 \pm 0.334$ | $68.498 \pm 0.976$ |
| GIN | $0.526 \pm 0.051$ | $96.485 \pm 0.252$ | $55.255 \pm 1.527$ | $85.387 \pm 0.136$ | $64.716 \pm 1.553$ |
| GAT | $0.384 \pm 0.007$ | $95.535 \pm 0.205$ | $64.223 \pm 0.455$ | $78.271 \pm 0.186$ | $70.587 \pm 0.447$ |
| GatedGCN | $0.282 \pm 0.015$ | $97.340 \pm 0.143$ | $67.312 \pm 0.311$ | $85.568 \pm 0.088$ | $73.840 \pm 0.326$ |
| GatedGCN-LSPE | $0.090 \pm 0.001$ | – | – | – | – |
| PNA | $0.188 \pm 0.004$ | $97.94 \pm 0.12$ | $70.35 \pm 0.63$ | – | – |
| DGN | $0.168 \pm 0.003$ | – | $72.838 \pm 0.417$ | $86.680 \pm 0.034$ | – |
| GSN | $0.101 \pm 0.010$ | – | – | – | – |
| CIN | $0.079 \pm 0.006$ | – | – | – | – |
| CRaW1 | $0.085 \pm 0.004$ | $97.944 \pm 0.050$ | $69.013 \pm 0.259$ | – | – |
| GIN-AK+ | $0.080 \pm 0.001$ | – | $72.19 \pm 0.13$ | $86.850 \pm 0.057$ | – |
| SAN | $0.139 \pm 0.006$ | – | – | $86.581 \pm 0.037$ | $76.691 \pm 0.65$ |
| Graphormer | $0.122 \pm 0.006$ | – | – | – | – |
| K-Subgraph SAT | $0.094 \pm 0.008$ | – | – | $86.848 \pm 0.037$ | $77.856 \pm 0.104$ |
| EGT | $0.108 \pm 0.009$ | $98.173 \pm 0.087$ | $68.702 \pm 0.409$ | $86.821 \pm 0.020$ | $79.232 \pm 0.348$ |
| Graphormer-URPE | $0.086 \pm 0.007$ | – | – | – | – |
| Graphormer-GD | $0.081 \pm 0.009$ | – | – | – | – |
| GPS | $0.070 \pm 0.004$ | $98.051 \pm 0.126$ | $72.298 \pm 0.356$ | $86.685 \pm 0.059$ | $78.016 \pm 0.180$ |
| GRIT | $0.059 \pm 0.002$ | $98.108 \pm 0.111$ | $76.468 \pm 0.881$ | $87.196 \pm 0.076$ | $80.026 \pm 0.277$ |
| GPNN-PE | $0.063 \pm 0.002$ | $98.165 \pm 0.077$ | $75.505 \pm 0.642$ | $87.083 \pm 0.035$ | $78.878 \pm 0.152$ |
| GPNN-PE (share-$P$) | $0.064 \pm 0.002$ | $98.018 \pm 0.024$ | $75.050, 0.282$ | $87.045 \pm 0.032$ | $78.830 \pm 0.127$ |
| GPNN-PE (1-head) | $0.066 \pm 0.005$ | $97.560 \pm 0.090$ | $72.042 \pm 0.714$ | $86.965 \pm 0.043$ | $78.373 \pm 0.212$ |

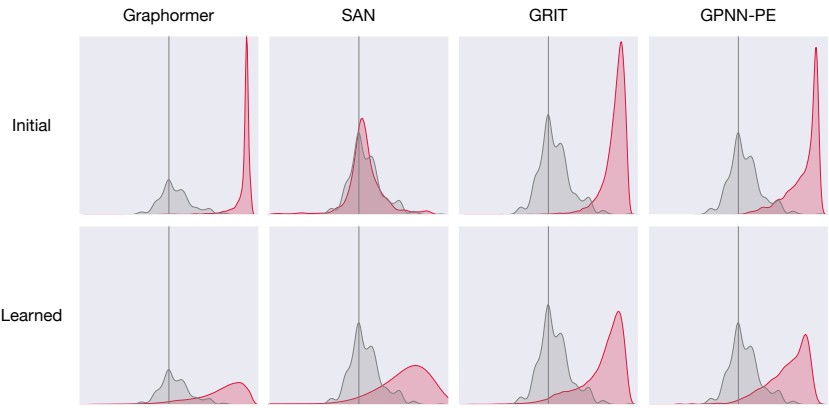

Figure 2: The KDE plot of CURC for initial/learned propagation graphs (red) and the input graphs (grey).

## 5.3 ANALYZE PROPAGATION GRAPHS WITH CURC

### 5.3.1 CURC DISTRIBUTION SHIFT

We analyze the learning dynamic of the propagation graphs via the CURC distribution. Comparing Graphormer (Ying et al., 2021), SANs (Kreuzer et al., 2021a), GRIT (Ma et al., 2023) and GPNN-PE on sampled graphs from ZINC datasets, the kernel density estimate (KDE) plots of CURC for the initial and learned propagation graphs are shown in Fig. 2. For pure transformers like Graphormer and GRIT, as well as GPNN-PE, the initial propagation graphs resemble a smoothed complete graph resulting in a right-skewed CURC distribution with nearly all positive curvatures. After epochs of training, the learned propagation graphs recover several negative curvatures, indicating the geometric patterns learned from the graphs. As a hybrid graph transformer, SAN owes an initial CURC

distribution close to input graphs. However, after epochs of training, SAN acquires a CURC distribution close to other graph transformers. One potential explanation is that there might exist an optical CURC distribution for learning graphs, balancing the risk of over-squashing (many edges of negative curvatures) and the danger of over-smoothing (many edges of large positive curvatures).

# 6 RELATED WORK

**Graph Rewiring** has been introduced to combat the over-squashing phenomenon in MPNN. Topping et al. (2022) proposed an iterative graph rewiring algorithm based on Balanced Forman Ricci curvature to mitigate the effect of negatively-curved edges on bottlenecks. Gutteridge et al. (2023) introduced a delayed message passing mechanism, which dynamically performs rewiring on graphs in the form of $k$-hop skip connections. Brüel-Gabrielsson et al. (2022) employed the strategy to rewire the node to all the other nodes in a receptive field and use positional encoding to describe the original graph structure.

**Graph Positional Encoding** is first proposed to enhance the performance of MPNNs (Zhang et al., 2021; Lim et al., 2023; Wang et al., 2022; Dwivedi et al., 2021; Bouritsas et al., 2022a). and becomes a crucial component in graph transformers (Dwivedi & Bresson, 2021; Kreuzer et al., 2021a; Rampášek et al., 2022; Ying et al., 2021; Zhang et al., 2023b; Ma et al., 2023).

**Expressivity** is an eternal topic of graph neural networks (GNNs) research. Xu et al. (2019) first points out the expressiveness limitation of MPNNs bounded by 1-WL algorithm on the graph isomorphism test. Follow-up works have attempted to breakthrough via higher-order-GNNs (Morris et al., 2019a; Bodnar et al., 2022; 2021), structural and positional encoding (Bouritsas et al., 2022a; Zhang et al., 2023b) and subgraph aggregation (Bevilacqua et al., 2022; Zhou et al., 2023).

**Ollivier-Ricci curvature.** There are various works investigating the sole mathematical properties of OR-curvatures proposed by Ollivier (2009). Jost & Liu (2014) reveals OR-curvature's connection with local clustering coefficient on undirected graphs where Topping et al. (2022) utilized to obtain a lower bound estimation for performing graph rewiring. Without restricting it to an unweighted-undirected graph, Bai et al. (2020) extends OR-curvature to weighted graphs by using the weighted graph Laplacian and carries the discussion to continuous-time Ollivier-Ricci flow. Ozawa et al. (2020) extends OR-curvature to strongly-connected weighted-directed graph using Perron measure with canonical shortest distance function, and we extend the curvature to weighted distance function to obtain CURC.

# 7 CONCLUSION AND FUTURE WORK

In this work, we introduce Generalized Propagation Neural Networks (GPNNs), a framework that formulates GTs along with various GNN families in a unified way. By studying the expressiveness of GPNNs, we reveal that the expressiveness of GPNNs primarily depends on the adjacency function. Providing a useful tool for future exploration in this direction. The introduction of Continuous Unified Ricci Curvature (CURC) enables in-depth analysis of propagation graphs, and empirical studies challenge the common belief regarding the benefits of global self-attention in graph transformers. For future work, exploring the implications of these findings, further refining the GPNN framework, and conducting additional empirical studies could yield more insights and advancements in the field of graph neural networks.

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

# Appendices

# A BACKGROUND

## A.1 PERMUTATION EQUIVARIANCE

Learning on graphs involves modeling a function that exhibits permutation equivariance on $\mathcal{G}$. Formally, for each element $\sigma$ in the symmetry group $\mathrm{S}_n$, an equivariant function $f : \mathbb{R}^{n^p} \mapsto \mathbb{R}^{n^k}$ adheres to

$$f(\sigma \cdot \mathbf{X}) = \sigma \cdot f(\mathbf{X}) \tag{9}$$

Suppose $\boldsymbol{X} \in \mathbb{R}^{n^p}$ and $\cdot$ denotes the action of permutation on the tensor power space of $\mathbb{R}^n$. Note that whenever additional dimensions for attributes appear, e.g. $f : \mathbb{R}^{n^p \times d} \mapsto \mathbb{R}^{n^k \times d}$, the action of permutation does not affect $d$. In the case when $k = 0$, the trivial action can be omitted, yielding

$$f(\sigma \cdot \mathbf{X}) = f(\mathbf{X}) \tag{10}$$

This denotes a specific instance of permutation invariance.

## A.2 OLLIVIER-RICCI CURVATURE

In differential geometry, Ricci curvature is a fundamental concept related to volume growth and allowing to classify the local characteristic of the space (roughly, whether it is sphere- or hyperboloid-like). Curvature also determines the behavior of parallel lines (whether they converge or diverge), known as geodesic dispersion. Discrete curvatures are analogous constructions for graphs (or more generally, metric spaces) trying to mimic some properties of the continuous curvature.

Ollivier (2009) introduced a notion of curvature for metric spaces that measures the Wasserstein distance between Markov chains, i.e. random walks, defined on two nodes. Let $\mathcal{G}$ be a graph with a distance metric $d_{\mathcal{G}}$, and $\mu_v$ be a probability measure on $\mathcal{G}$ for node $v \in \mathcal{V}$. The Ollivier–Ricci curvature of any pair $\{(i,j)|(x,y) \in \mathcal{V}^2, x \neq y\}$ is defined as

$$\kappa_{\mathrm{OR}}(x,y) := 1 - \frac{1}{d_G(x,y)} W_1\left(\mu_x, \mu_y\right), \tag{11}$$

where $W_1$ refers to the first Wasserstein distance between $\mu_i$ and $\mu_j$.

Ollivier-ricci curvature is the most prominent discrete curvature on metric spaces which quantifies how the geometry of the manifold deviates from flat (Euclidean) space in terms of the metric structure. Other choices of discrete curvatures include the **Forman–Ricci Curvature**

$$\kappa_{\mathrm{FR}}(x,y) := 4 - d_x - d_y + 3\left|\#_\triangle\right|$$

and the **Resistance Curvature**

$$\kappa_{\mathrm{R}}(x,y) := \frac{2\left(p_x + p_y\right)}{R_{xy}}.$$

Specifically, we build on our the framework of OR-curvature, due to its intrinsic relation with MPNN and its geometric and spectral properties relating to graph structure.

# B THEORETICAL DETAILS

## B.1 PROPERTIES OF CONTINUOUS UNIFIED RICCI CURVATURE

Kantorovich-Rubinstein duality is an important result in the field of optimal transportation, which establishes the connection between transportation problem and Linear programming problem. While the most used form of duality is stated in the context of Polish metric space, in the setting up of our $\kappa_{\mathrm{GOR}}$, the distance function as weighted shortest distance is not necessarily symmetric, which fails to define a metric space. Luckily, the duality still holds under weaker assumption and we will give a short proof of Kantorovich-Rubinstein duality for the sake of completeness.

**Definition B.1.** Let $d : \mathcal{X} \times \mathcal{X} \mapsto \mathbb{R}^{\geq 0}$ be an asymmetric distance function on $\mathcal{X}$, we say $f : \mathcal{X} \mapsto \mathbb{R}$ is L-Lipschitz w.r.t. $d$ if
$$\forall x, y \in \mathcal{X}, f(y) - f(x) \leq Ld(x, y).$$

**Definition B.2.** Let $\mathcal{X}$ be a finite set and $\mu : \mathcal{X} \mapsto [0, 1]$ be a corresponding probability measure, we define the *support* of $\mu$ by
$$\mathrm{supp}(\mu) = \{x \in \mathcal{X} : \mu(x) > 0\}$$

**Definition B.3.** Suppose $\mu$ and $\nu$ to be two probability distribution on finite sets $\mathcal{X}$ and $\mathcal{Y}$ respectively. Let $\Pi(\mu, \nu)$ denotes the set of *couplings* between $\mu$ and $\nu$. We say $\pi : \mathcal{X} \times \mathcal{Y} \mapsto [0, 1] \in \Pi(\mu, \nu)$ is a *coupling* if
$$\sum_{y \in V} \pi(x, y) = \mu(x), \quad \sum_{x \in V} \pi(x, y) = \nu(y).$$

**Definition B.4.** Let $\mathcal{X}$ and $\mathcal{Y}$ be two sets and $c : \mathcal{X} \times \mathcal{Y} \mapsto \mathbb{R} \cup \{+\infty\}$. A function $f : \mathcal{X} \mapsto \mathbb{R} \cup \{+\infty\}$ is *c-convex* if it is not identically $+\infty$, and there exists $\psi : \mathcal{Y} \mapsto \mathbb{R} \cup \{+\infty\}$ such that
$$\forall x \in \mathcal{X}, f(x) = \sup_{y \in \mathcal{Y}} (\psi(y) - c(x, y)).$$

Then its corresponding *c-transform* is the function $\psi^c$ defined by
$$\forall y \in \mathcal{Y}, f^c(y) = \inf_{x \in \mathcal{X}} (\psi(x) + c(x, y)).$$

**Lemma B.1.** *Let $f : \mathcal{X} \mapsto \mathbb{R}$ be a function defined on a set $\mathcal{X}$. Let $d : \mathcal{X} \times \mathcal{X} \mapsto \mathbb{R}^{\geq 0}$ to be a distance function on $\mathcal{X}$ that satisfies the following properties:*

- $\forall x \in \mathcal{X}, d(x, x) = 0$

- $\forall x \neq y \in \mathcal{X}, d(x, y) > 0$

- $\forall x, y, z \in \mathcal{X}, d(x, z) + d(z, y) \geq d(x, y)$

*Then function $f$ is $d$-convex $\iff f$ is 1-Lipschitz w.r.t. distance function d.*

*Proof.* We first suppose $f$ is $d$-convex, we want to show that $\forall x, y \in \mathcal{X}, f(x) - f(y) \leq d(y, x)$. By the definition of *c-convex*, $\exists$ function $\psi : \mathcal{X} \mapsto \mathbb{R}$, such that $f(x) = \sup_{z \in \mathcal{X}} [\psi(z) - d(x, z)]$ and $f(x) = \sup_{z \in \mathcal{X}} [\psi(z) - d(y, z)]$. Suppose $z_0 = \arg\sup_{z \in \mathcal{X}} [\psi(z) - d(x, z)]$
$$\begin{aligned} f(x) - f(y) &= \sup_{z \in \mathcal{X}} [\psi(z) - d(x, z)] - \sup_{z \in \mathcal{X}} [\psi(z) - d(y, z)] \\ &\leq [\psi(z_0) - d(x, z_0)] - [\psi(z_0) - d(y, z_0] \\ &= d(y, z_0) - d(x, z_0) \\ &\leq d(y, x) \text{ by triangular inequality} \end{aligned}$$

Now, suppose $f$ is 1-Lipschitz w.r.t. distance function $d$. We note that $f^c(y) = \inf_{x \in \mathcal{X}} [f(x) + d(x, y)]$. By 1-Lipschitz, we have that
$$\begin{aligned} f(y) - f(x) &\leq d(x, y) \\ \Rightarrow f(x) - f(y) &\geq -d(x, y) \\ \Rightarrow f(x) &\geq f(y) - d(x, y) \\ \Rightarrow f(x) &\geq \sup_{y \in \mathcal{X}} [f(y) - d(x, y)] \end{aligned}$$

by taking $x = y$ in the supremum and $d(x, x) = 0$, we have the following equality:

$$f(x) = \sup_{y \in \mathcal{X}} [f(y) - d(x, y)]$$

By the exact same argument, we can derive a bonus property that

$$f^c(y) = \inf_{x \in \mathcal{X}} [f(x) + d(x, y)] = f(y).$$

Therefore, on a set $\mathcal{X}$ with asymmetric distance function that satisfies triangle inequality, function $f$ is *d-convexity* $\iff$ $f$ is 1-Lipschitz w.r.t. distance function $d$. And its *c-transform* is itself. $\square$

**Theorem B.2.** *Let $\mathcal{V}$ be a discrete finite set equipped with a possibly asymmetric distance function $d : \mathcal{V} \times \mathcal{V} \mapsto \mathbb{R}^{\geq 0}$ that satisfies triangle inequality. Suppose $\mu$ and $\nu$ to be two probability measure on $\mathcal{V}$. Then we have the Kantorovich–Rubinstein duality:*

$$\inf_{\pi \in \Pi(\mu, \nu)} \sum_{x, y \in \mathcal{V}} d(x, y) \pi(x, y) = \sup_{f \in \mathrm{Lip}(1)} \sum_{x \in \mathcal{V}} f(x)(\nu(x) - \mu(x)), \tag{1}$$

*where $\Pi(\mu, \nu)$ denotes the coupling between probability measure $\mu$ and $\nu$. $f \in \mathrm{Lip}(1)$ denotes that $f : \mathcal{V} \mapsto \mathbb{R}$ is 1-Lipschitz w.r.t. distance function $d$.*

*Proof.* We prove equation 1 by first showing that

$$\inf_{\pi \in \Pi(\mu, \nu)} \sum_{x, y \in \mathcal{V}} d(x, y) \pi(x, y) \geq \sup_{f \in \mathrm{Lip}(1)} \sum_{x \in \mathcal{V}} f(x)(\nu(x) - \mu(x)). \tag{2}$$

Then we provide a specific construction of 1-Lipschitz function $f : \mathcal{V} \mapsto \mathbb{R}$ showing the converse,

$$\inf_{\pi \in \Pi(\mu, \nu)} \sum_{x, y \in \mathcal{V}} d(x, y) \pi(x, y) \leq \sup_{f \in \mathrm{Lip}(1)} \sum_{x \in \mathcal{V}} f(x)(\nu(x) - \mu(x)). \tag{3}$$

Firstly, take arbitrary $\pi \in \Pi(\mu, \nu)$ and $f$ 1-Lipschitz, we have the following algebraic property:

$$\sum_{x \in \mathcal{V}} f(x)(\nu(x) - \mu(x)) = \sum_{x, y \in \mathcal{V}} f(x) \pi(x, y) - \sum_{x, y \in \mathcal{V}} f(y) \pi(x, y)$$

$$= \sum_{x, y \in \mathcal{V}} [f(y) - f(x)] \pi(x, y)$$

$$\leq \sum_{x, y \in \mathcal{V}} d(x, y) \pi(x, y)$$

Therefore,

$$\inf_{\pi \in \Pi(\mu, \nu)} \sum_{x, y \in \mathcal{V}} d(x, y) \pi(x, y) \geq \sum_{x \in \mathcal{V}} f(x)(\nu(x) - \mu(x))$$

$$\Rightarrow \inf_{\pi \in \Pi(\mu, \nu)} \sum_{x, y \in \mathcal{V}} d(x, y) \pi(x, y) \geq \sup_{f \in \mathrm{Lip}(1)} \sum_{x \in \mathcal{V}} f(x)(\nu(x) - \mu(x)),$$

which is exactly equation 2.

Secondly, we construct a function with the help of *c-convexity* introduced in definition B.4. For a fixed $m \in \mathbb{N}$, we can pick a sequence $(x_i, y_i)_{i=0}^m \in \mathrm{supp}(\pi)$. Note that this choice of $m$ is finite and $m \leq |\mathcal{V}|^2$. We construct our function $f$ by

$$f(x) := \sup_{m \in \mathbb{N}} \sup_{(x_i, y_i)_{i=0}^m} \{ \sum_{i=0}^{m-1} [d(x_i, y_i) - d(x_{i+1}, y_i)] + [d(x_m, y_m) - d(x, y_m)] \}. \tag{4}$$

We now want to show that $f^c(y) - f(x) = d(x, y)$ almost surely for $(x, y) \in \mathrm{supp}(\pi)$. Note that

$$f^c(y) = \inf_{x \in \mathcal{X}} [f(x) + d(x, y)]$$

$$\Rightarrow f^c(y) \leq f(x) + d(x, y)$$

$$\Rightarrow f^c(y) - f(x) \leq d(x, y).$$

By lemma B.1, we have that $f^c(y) = f(y)$, which implies that this choice of $f$ guarantees 1-Lipschitz.

Suppose $(x, y) \in \text{supp}(\pi)$, and in the choice of sequence $(x_i, y_i)_{i=0}^{m}$ we can let $(x_m, y_m) := (x, y)$. Therefore,

$$
\begin{aligned}
f(z) \geq \sup_{m \in \mathbb{N}} \sup_{(x_i,y_i)_{i=0}^{m}} \Big\{ &\sum_{i=0}^{m-2} [d(x_i, y_i) - d(x_{i+1}, y_i)] + [d(x_{m-1}, y_{m-1}) - d(x, y_{m-1})] \\
&+ [d(x, y) - d(z, y)] \Big\} \\
=& f(x) + d(x, y) - d(z, y).
\end{aligned}
$$

The last equality comes from the fact that in the definition of $f$, taking supremum over $m$ or $m - 1$ does not matter. Hence,

$$
\begin{aligned}
&f(z) + d(z, y) \geq f(x) + d(x, y) \\
\Rightarrow &\inf_{z \in \mathcal{V}} [f(z) + d(z, y)] \geq f(x) + d(x, y) \\
\Rightarrow &f^c(y) \geq f(x) + d(x, y) \\
\Rightarrow &f^c(y) - f(x) \geq d(x, y)
\end{aligned}
$$

Therefore, we have that $\forall (x, y) \in \text{supp}(\pi), f^c(y) - f(x) = d(x, y) \iff f(y) - f(x) = d(x, y)$ by using lemma B.1 again. For clarity, we will denote this choice of $f$ to be $f^*$. Using this result, we have $\forall \pi \in \Pi(\mu, \nu)$,

$$
\begin{aligned}
\sup_{f \in \text{Lip}(1)} \sum_{x \in \mathcal{V}} f(x)(\nu(x) - \mu(x)) &\geq \sum_{x \in \mathcal{V}} f^*(x)(\nu(x) - \mu(x)) \\
&= \sum_{x,y \in \mathcal{V}} [f^*(y) - f^*(x)] \pi(x, y) \\
&= \sum_{x,y \in \mathcal{V}} d(x, y) \pi(x, y) \\
&\geq \inf_{\pi \in \Pi(\mu,\nu)} \sum_{x,y \in \mathcal{V}} d(x, y) \pi(x, y)
\end{aligned}
$$

Therefore, we have both equations 2 and 3 hold, hence we have proven the Kantorovich–Rubinstein duality under the weaker assumption that distance function $d : \mathcal{V} \times \mathcal{V} \mapsto \mathbb{R}^{\geq 0}$ is not necessarily symmetric. $\square$

**Corollary B.3.** *Suppose $\mathcal{V}$ to be a discrete finite set equipped with a possibly asymmetric distance function $d : \mathcal{V} \times \mathcal{V} \mapsto \mathbb{R}^{\geq 0}$ that satisfies triangle inequality. Suppose $\mu$ and $\nu$ to be two probability measure on $\mathcal{V}$ with support on $\{x_1, x_2, \ldots, x_n\}$ and $\{y_1, y_2, \ldots, y_m\}$ respectively. Then solving the Wasserstein distance $\mathcal{W}_1(\mu, \nu) = \inf_{\pi \in \Pi(\mu,\nu)} \sum_{x,y \in \mathcal{V}} d(x, y) \pi(x, y)$ is equivalent with*

$$
W_1(\mu, \nu) = \sup_{f \in \text{Lip}(1)} \left\{ \sum_i f(x_i) \nu(x_i) - \sum_j f(y_j) \mu(y_j) \right\}, \tag{5}
$$

*which is further equivalent to the following Linear programming problem*

$$
W_1(\mu, \nu) = \sup_{Af \preceq c} m^T f, \tag{6}
$$

*with the following construction of matrix and vectors*

$m := (\mu(x_1), \ldots, \mu(x_n), \nu(y_1), \ldots, \nu(y_m))^T \in \mathbb{R}^{n+m},$

$\phi := (f(x_1), \ldots, f(x_n), -f(y_1), \ldots, -f(y_m))^T \in \mathbb{R}^{n+m},$

$c := (d(x_1, y_1), \ldots, d(x_1, y_m), d(x_2, y_1), \ldots, d(x_n, y_1), \ldots, d(x_n, y_m),$

$d(y_1, x_1), \ldots, d(y_m, x_1), d(y_1, x_2), \ldots, d(y_1, x_n), \ldots, d(y_m, x_n))^T \in \mathbb{R}^{nm},$

$$
A := \begin{pmatrix} A_1 \\ A_2 \end{pmatrix}, \text{where } A_1 := \begin{pmatrix} a_1 & I_m \\ a_2 & I_m \\ \vdots & \vdots \\ a_n & I_m \end{pmatrix}, A_2 := -A_1, a_i \in \mathbb{R}^{m \times n} \text{ with all ones i-th column.}
$$

**Lemma B.4.** *We say a matrix $A \in \mathbb{R}^{n \times n}$ is regular if for some $k \geq 1$, $A^k > 0$. Or equivalently, matrix $A$ has non-negative entries and is strongly connected in our context. Then by Perron-Frobenius theorem:*

- *There exists a unique positive unit left eigenvector $\boldsymbol{v}_{pf}$ of $A$ called Perron-Frobenius left eigenvector, whose corresponding eigenvalue $\lambda_{pf}$ is real and has the largest norm among all eigenvalues.*

- *Let $\lambda_{pf}$ be the corresponding eigenvalue of $\boldsymbol{v}_{pf}$, for any other eigen*

- *$\lambda_{pf}$ is simple, i.e. has multiplicity one*

*Proof.* Perron-Frobenius theorem is well-known in the field of linear algebra, and has different forms on non-negative matrices, non-negative regular matricesa and postivie matrices. We only need it for non-negative regular matrices. □

**Lemma B.5.** *Let $A(t)$ be a differentiable matrix-valued function of $t$, $a(t)$ an eigenvalue of $A(t)$ of multiplicity one. Then we can choose an eigenvector $h(t)$ of $A(t)$ pertaining to the eigenvalue $a(t)$ to depend differentiably on $t$.*

*Proof.* For the purpose of our proof, we only need continuity of $h(t)$ on $t$, but we present this stronger statement, cf. Theorem 8, p130 in (Lax, 2007). □

**Lemma B.6.** *Suppose $\omega \in \mathbb{R}^{n \times n}$ is a non-negative regular matrix. Then its Perron-Frobenius left eigenvector $\boldsymbol{v}_{pf}$ depend continuously on the $\omega$ w.r.t. **small** perturbation $\varepsilon$ entry-wise, where $W$ is still a non-negative regular matrix after the perturbation.*

*Proof.* Let $E_{ij}$ denotes a matrix with zero entries except for entry $(i,j)$. Let $A(\varepsilon) := \omega + E_{ij}\varepsilon$, which is obviously a matrix-valued function differentiable w.r.t. $\varepsilon$. Suppose $|\varepsilon|$ is small such that we are only dealing with non-negative regular $A(\varepsilon)$. Therefore by Perron-Frobenius theorem B.4, there exists $\lambda_{pf}(\varepsilon)$ and $\mathbf{v}_{pf}(\varepsilon)$ for $A(\varepsilon)$, which has multiplicity 1. Therefore, by lemma B.5, we have that $\mathbf{v}_{pf}(\varepsilon)$ continuously depends on $\varepsilon$. Note that this eigenvector unnecessarily has unit length. But fortunately, the 2-norm of a positive continuous vector function is also continuous w.r.t. $\varepsilon$, we have that the Perron-measure $\mathfrak{m} := \frac{\mathbf{v}_{pf}(\varepsilon)}{\|\mathbf{v}_{pf}(\varepsilon)\|}$ is continuous w.r.t. $\varepsilon$ element-wise. □

**Lemma B.7.** *The mean transition probability $\mu_x$ for node $x$ defined in equation 7 is continuous w.r.t. weight matrix $\omega$ entry-wise with $\omega$ still non-negative and regular.*

*Proof.* By lemma B.6, we have that the Perron-measure $\mathfrak{m}$ is a continuous function w.r.t. $\omega$ entry-wise. While $\mathfrak{m}$ is a vector-valued function of $\omega$, each entry can be viewed as a continuous function of $\omega$ entry-wise. Since $\forall\, x \in \mathcal{V}$, $\mathfrak{m}(x) > 0$, we have $\forall x, y \in \mathcal{V}$, $\frac{\mathfrak{m}(y)}{\mathfrak{m}(x)}$ is continuous w.r.t. $\omega$ entry-wise. Now we consider normalized weight $W$. WLOG, we suppose a perturbation of $\delta$ on $\omega$ in entry $(i,j)$, which only influences the $i$-th row of $W$. Denote this perturbed normalized weight matrix by $W^*$, and we have that

$$W^*(x,y) = \begin{cases} \frac{W(x,y)}{1+\delta} & \text{if } x = i, y \neq j \\ \frac{W(x,y)+\delta}{1+\delta} & \text{if } x = i, y = j \\ W(x,y) & \text{if } x \neq i \end{cases}$$

Therefore if we choose to perturb the $(i,j)$ entry of $W$, the value of $W(x,y)$ is indifferent to this entry, hence continuous. For $W(i,y)$ where $y \neq j$, we pick $\varepsilon > 0$. By choosing $\delta \leq \frac{\varepsilon}{W(i,y)-\varepsilon}$, we ensure $W(x,y) - \frac{W(x,y)}{1+\varepsilon} \leq \varepsilon$, hence $W(i,y)$ is continuous w.r.t. $\omega$ entry-wise. For $W(i,j)$, we also pick $\varepsilon > 0$, and we choose $\delta \leq \frac{\varepsilon}{1-W(i,j)-\varepsilon}$ to get the entry-wise continuity. Therefore, $\forall x \in \mathcal{V}$, the mean transition measure $\mu_x$

$$\mu_x(y) := \frac{1}{2}[W(x,y) + \frac{\mathfrak{m}(y)}{\mathfrak{m}(x)}W(y,x)],$$

is a continuous function w.r.t. $\omega$ entry-wise, as summation and product of two continuous function is still continuous. □

**Lemma B.8.** *Consider the convex optimization problem with a valid solution:*

$$\mathcal{M} := \inf_{Af \preceq c(t)} m^T f, \tag{7}$$

*where $A \in \mathbb{R}^{n \times m}, f \in \mathbb{R}^{m \times 1}, m \in \mathbb{R}^{n \times 1}$, and c(t): $\mathbb{R} \mapsto \mathbb{R}^{n \times 1}$ being a vector-valued function that depends continuously on $t \in \mathbb{R}$. We claim that $\mathcal{M}$ also depends continuously on $t$.*

*Proof.* Note that equation 7 is a convex optimization problem and admits a feasible solution. Therefore the optimization problem admits strong duality from the Slater's condition and is equivalent to the following dual problem:

$$\text{maximize} \quad -c(t)^T g$$
$$\text{subject to} \quad A^T g + m = 0, \quad g \succeq 0.$$

In particular, the maximization problem has $\mathcal{M}$ as the optimal value. From the strong duality of convex optimization problem, we move the continuous function $c(t)$ from the constraint to the objective. Note that $-c(t)^T g$ is a continuous function of $t$, therefore the supremum over $-c(t)^T g$ is also a continuous function. $\qquad\square$

**Lemma B.9.** *Consider the convex optimization problem with a valid solution:*

$$\mathcal{M} := \inf_{Af \preceq c(t)} m(t)^T f,$$

*where $A \in \mathbb{R}^{n \times m}, f \in \mathbb{R}^{m \times 1}$ and $m(t) : \mathbb{R} \mapsto \mathbb{R}^{n \times 1}$ and $c(t) : \mathbb{R} \mapsto \mathbb{R}^{n \times 1}$ being vector-valued functions that depend continuously on $t \in \mathbb{R}$. We claim that $\mathcal{M}$ depends continuously on $t$.*

*Proof.* We extend the lemma to the case where the objective is also a continuous function of $t$. Let $\delta > 0$ be a small perturbation, and $\mathcal{M}^* = \inf_{Af \preceq c(t+\delta)} m(t+\delta)^T f$. Therefore, we have that

$$|\mathcal{M} - \mathcal{M}^*| = |\inf_{Af \preceq c} m^T f - \inf_{Af \preceq c^*} m^{*T} f|$$
$$= |\inf_{Af \preceq c} m^T f - \inf_{Af \preceq c^*} m^T f + \inf_{Af \preceq c^*} m^T f - \inf_{Af \preceq c^*} m^{*T} f|$$
$$\leq |\inf_{Af \preceq c} m^T f - \inf_{Af \preceq c^*} m^T f| + |\inf_{Af \preceq c^*} m^T f - \inf_{Af \preceq c^*} m^{*T} f|$$

By lemma B.8 and continuity of supremum over continuous function, $\mathcal{M}$ depends continuously on variable $t$. $\qquad\square$

The following is the proof of three important properties of CURC, the first one and the third one are natural results from the construction of $\kappa_{\text{CURC}}$, but we stress that the second note on continuity of $\kappa_{\text{CURC}}$ is non-trivial and new.

**Proposition B.10.** *We have the following properties for $\kappa_{\text{GOR}}$:*

- *For connected unweighted-undirected graph $\mathcal{G} = (\mathcal{V}, \mathcal{E})$, for any pair of nodes $x, y \in \mathcal{V}$, we have $\kappa_{\text{CURC}}(x, y) = \kappa_{\text{OR}}(x, y)$.*

- *If we perceive $\kappa_{\text{CURC}}(x, y)$ as a function of $\omega \in \mathbb{R}^{n \times n}$, then $\kappa_{\text{CURC}}(x, y)$ is continuous w.r.t. $\omega$ entry-wise.*

- *For strongly-connected weighted-directed graph $\mathcal{G} = (\mathcal{V}, \mathcal{E}, \omega)$, when all edge weights $\omega$ are scaled by an arbitrary positive constant $\lambda$, the value of $\kappa_{\text{CURC}}(x, y)$ for any $x, y \in \mathcal{V}$ is unchanged.*

*Proof.* We prove the properties by sequential order.

1. **Proof of the first property**
   Suppose $A$ to be the adjacency matrix (binary) of unweighted-undirected graph $\mathcal{G} = (\mathcal{V}, \mathcal{E})$ and we are interested in $\kappa_{CURC}$ for $x, y \in \mathcal{V}$. Let $\mathcal{M} :=$ the length of the longest shortest path. Intuitively, by the definition of the $\varepsilon$-*masked* **CURC**, we may pick $\varepsilon$ sufficiently small,

such that none of the "virtual edges" masked as edge length $\frac{1}{\varepsilon}$ will be picked in calculating the weighted shortest distance. By picking $\epsilon < \frac{1}{\mathcal{M}}$, the "virtual edges" have length even more than the longest distance in $\mathcal{G}$, resulting $\forall x, y \in \mathcal{V}$, $d(x, y)$ is independent of these "virtual edges".

When $\mathcal{G}$ is unweighted-undirected, the adjacency matrix $A$ is symmetric. Therefore the Perron-measure $\mathfrak{m}(x)$ for each node $x$ is proportional to the inverse degree $\frac{1}{d_x}$. By direct calculation, we have $W(x, y) = \frac{1}{d_x}$ and $W(y, x) = \frac{1}{d_y}$. Therefore, the mean transition distribution

$$\mu_x(y) = \frac{1}{2}\left[\frac{1}{d_x} + \frac{d_y}{d_x \times d_y}\right] = \frac{1}{d_x},$$

which is equivalent to the initial mass placement in the construction of $\kappa_{OR}$.

Since the initial mass distribution according for $\kappa_{CURC}$ and $\kappa_{OR}$ is the same and we choose $\epsilon$ sufficiently small, namely when $\epsilon < \frac{1}{\mathcal{M}}$, the masked and unmasked Wasserstein distances are equal: $\mathcal{W}_1^\varepsilon(\mu_x, \mu_y) = \mathcal{W}_1(\mu_x, \mu_y)$.

It is straightforward that the shortest weighted distance $d$ is positively related to $\varepsilon$. Hence $\kappa_{CURC}^\varepsilon$ is a decreasing function in $\varepsilon$. But for a fixed graph $\mathcal{G}$, $\kappa_{CURC}^\varepsilon$ is invariant when $\varepsilon < \frac{1}{\mathcal{M}}$. Therefore the limit of $\kappa_{CURC}^\varepsilon$ indeed tends to $\kappa_{CURC}$ as $\varepsilon \to 0$.

2. **Proof of the second property**

To prove the entry-wise continuity of **CURC** w.r.t. weight matrix $\omega$ under the assumption that perturbation ensures $\omega$ to be non-negative and regular, we use the supremum form of Wasserstein distance from the Kantorovich–Rubinstein duality:

$$\mathcal{W}_1^\varepsilon(\mu_x, \mu_y) = \sup_{f \in \text{Lip}(1)} \sum_{z \in \mathcal{V}} f(z)\left(\mu_y(z) - \mu_x(z)\right), \qquad (8)$$

where $\forall x, y \in V, f(y) - f(x) \leq d^\varepsilon(x, y)$ as $f \in \text{Lip}(1)$. We may exploit the limit definition of $\mathcal{W}_1^\varepsilon$ and take $\varepsilon$ sufficiently small so that $d^\epsilon$ and $\mathcal{W}_1^\varepsilon$ is independent of $\varepsilon$, hence we may abuse the notation, denote $d^\varepsilon(x, y)$ as $d(x, y)$ and $\mathcal{W}_1^\varepsilon(\mu_x, \mu_y)$ as $\mathcal{W}_1(\mu_x, \mu_y)$ for $x, y \in \mathcal{V}$. We first show that $\forall x \neq y \in \mathcal{V}$, $d(x, y)$ is continuous entry-wise w.r.t. $\omega$. Let $\mathcal{M}$ be the diameter w.r.t. edge length as inverse edge weights $\frac{1}{\omega}$, and for edge with $0$ weight we treat the edge length as $+\infty$. Suppose there is a small perturbation of $\delta$ on an arbitrary entry of $\omega$ after which the weight matrix is still non-negative regular, we denote the new weight matrix as $\omega^\delta$ and distance function as $d^\delta$ on this perturbed matrix. WLOG, we assume the perturbation is smaller than the smallest positive entry and we let $\omega^{min}$ to be this minimal positive entry of $\omega$. With this small perturbation, we have

$$|d(x, y) - d^\delta(x, y)| \leq \frac{1}{\omega^{min} - \delta} - \frac{1}{\omega^{min}}.$$

Therefore, $\forall \varepsilon_0$, let $\delta \leq \frac{\omega^2 \varepsilon_0}{1 + \omega \varepsilon_0}$, we have $|d(x, y) - d^\delta(x, y)| \leq \varepsilon_0$, hence $d(x, y)$ is continuous entry-wise w.r.t. $\omega$.

We will prove the continuity of

$$\frac{\mathcal{W}_1(\mu_x, \mu_y)}{d(x, y)} = \sup_{f \in \text{Lip}(1)} \sum_{z \in \mathcal{V}} f(z) \frac{\mu_y(z) - \mu_x(z)}{d(x, y)},$$

which is sufficient for the overall continuity of **CURC**. By corollary B.3, we have that

$$\mathcal{W}_1(\mu_x, \mu_y) = \sup_{Af \preceq c} m^T f.$$

By previous argument, $m$ as a function of mean transition measure $\mu$ and $c$ as a function of distance function $d$ are both continous w.r.t. $\omega$ entry-wise. Therefore, $\mathcal{W}_1(\mu_x, \mu_y)$ is indeed a continuous function w.r.t. $\omega$ entry-wise by lemma B.9. Since $d(x, y)$ is positive and continuous entry-wise, we conclude that $\kappa_{CURC}$ is continuous entry-wise w.r.t. $\omega$.

3. **Proof of the third property**

The third property is straightforward from the construction of the edge length function $l^\varepsilon$. Suppose $\mathcal{G}(\mathcal{V}, \mathcal{E}, \omega)$ to be the unscaled strongly-connected weighted-directed graph and $\mathcal{G}^*(\mathcal{V}, \mathcal{E}, \omega^*)$ be the scaled one, where $\omega^* = \lambda\omega$. Let $\mathcal{M}$ and $\mathcal{M}^*$ be the diameter of $\mathcal{G}$ and

$\mathcal{G}^*$ respectively w.r.t. the inverse edge weight. Note we are not considering the $\varepsilon$-masked edge weight in the sense that for non-existent edges, the edge length is $+\infty$. $\mathcal{M}$ and $\mathcal{M}^*$ are well-defined due to strongly-connectivity.

After a scaling of $\lambda$, the eigenvalues of $\omega$ and $\omega^*$ differ by a factor of $\lambda$. By Perron-Frobenius theorem, the corresponding Perron-measure $\mathfrak{m}$ and $\mathfrak{m}^*$ are the same. Similarly, it is straightforward the weighted graph laplacian matrices $W$ and $W^*$ are also the same. Therefore the initial measure $\mu$ of $\mathcal{G}$ coincides with $\mu^*$ of $\mathcal{G}^*$.

Since $\lambda > 0$, we can find $\epsilon$ small enough, so that $\frac{1}{\epsilon} < \min\{\mathcal{M}, \mathcal{M}^*\}$, ensuring the distance function $d^\varepsilon$ and $d^{\varepsilon*}$ are independent of $\varepsilon$. Hence, $\forall x \neq y \in \mathcal{V}$, $d^{\varepsilon*}(x, y) = \lambda d^\varepsilon(x, y)$ By Kantorovich–Rubinstein duality B.2, we have that

$$
\begin{aligned}
\mathcal{W}_1^{\varepsilon*}(x, y) &= \inf_{\pi \in \Pi(\mu_x, \mu_y)} \sum_{x, y \in V} d^*(x, y) \pi(x, y) \\
&= \inf_{\pi \in \Pi(\mu_x, \mu_y)} \sum_{x, y \in V} \lambda d(x, y) \pi(x, y) \\
&= \lambda \mathcal{W}_1^\varepsilon(x, y).
\end{aligned}
$$

Therefore,

$$
\begin{aligned}
\kappa_{CURC}^* &= \lim_{\varepsilon \to 0} \kappa_{CURC}^{\varepsilon*}(x, y) \\
&= \lim_{\varepsilon \to 0} (1 - \frac{\mathcal{W}_1^{\varepsilon*}(\mu_x, \mu_y)}{d^{\varepsilon*}(x, y)}) \\
&= \lim_{\varepsilon \to 0} (1 - \frac{\lambda \mathcal{W}_1^\varepsilon(\mu_x, \mu_y)}{\lambda d^\varepsilon(x, y)}) \\
&= \lim_{\varepsilon \to 0} (1 - \frac{\mathcal{W}_1^\varepsilon(\mu_x, \mu_y)}{d^\varepsilon(x, y)}) \\
&= \kappa_{CURC},
\end{aligned}
$$

as required. Therefore, our **CURC** is free of scale.

$\square$

## B.2 GEOMETRIC PROPERTY CONTINUOUS UNIFIED RICCI CURVATURE

To reveal the geometric connection between Continuous Unified Ricci Curvature and graphs, we introduce the concept of *Dirichlet isoperimetric constant* $\mathcal{I}_\mathcal{V}^D$, which is the analogy of the well-known cheeger constant in strongly-connected weighted-directed graphs. We perceive $\mathcal{I}_\mathcal{V}^D$ as a measure of bottlenecking on weighted graphs and state that when **CURC** has a lower bound $K$, $\mathcal{I}_\mathcal{V}^D$ has a lower bound that is positively related to $K$. Our proof draws its foundation from the derivation presented in (Ozawa et al., 2020), where they prove the result on strongly-connected weighted-directed graph with unit edge length. Few minor modifications are required to adapt the result to our scenario concerning weighted edge lengths. We will identify and elucidate the key elements that require clarification, while also presenting the results that remain unchanged.

**Definition B.5.** Let $\mathcal{G} = (\mathcal{V}, \mathcal{E}, \omega)$ be a finite strongly-connected weighted directd-graph and $f : \mathcal{V} \mapsto \mathbb{R}$. Suppose $\mu : \mathcal{V} \times \mathcal{V} \mapsto [0, 1]$ is a probability kernel satisfying $\sum_{y \in \mathcal{V}} \mu(x, y) = 1$ for all $x \in \mathcal{V}$. The *Chung Laplacian* $\mathcal{L}$ on function $f$ associated with $\mu$ is defined as

$$
\mathcal{L}f(x) := f(x) - \sum_{y \in \mathcal{V}} \mu(x, y) f(y).
$$

Let $d : \mathcal{V} \times \mathcal{V} \mapsto \mathbb{R}^{\geq 0}$ be a distance function (asymmetric). For each node $x \in \mathcal{V}$, the *asymptotic mean curvature* $\mathcal{H}_x$ is defined by

$$
\mathcal{H}_x := \mathcal{L}\rho_x(x),
$$

where $\rho_x : \mathcal{V} \to \mathbb{R}$ is the distance from $x$ defined as $\rho_x(y) := d(x, y)$. Note that with weighted shortest distance $d$, $\mathcal{H}_x \in (-\infty, 0)$.

For each node $x \in \mathcal{V}$, the $\mathrm{InRad}_x \mathcal{V}$ of $\mathcal{V}$ at node $x$ is defined by

$$
\mathrm{InRad}_x \mathcal{V} := \sup_{y \in \mathcal{V}} \rho_x(y),
$$

And for any $x \in \mathcal{V}$ and $R > 0$, we set $E_R(x) := \{y \in V \mid \rho_x(y) \geq R\}$.

**Definition B.6.** Let $\mathcal{G} = (\mathcal{V}, \mathcal{E}, \omega)$ be a finite strongly-connected weighted-directed graph, $\mu : \mathcal{V} \times \mathcal{V} \mapsto [0, 1]$ be the mean transition kernel and $\mathfrak{m} : \mathcal{V} \mapsto [0, 1]$ be the Perron-measure in definition 4.1. Based on the Perron-measure $\mathfrak{m}$, we define $\mathfrak{m} : \mathcal{V} \times \mathcal{V} \mapsto [0, 1]$ by

$$\mathfrak{m}(x, y) := \frac{1}{2}(\mathfrak{m}(x)\mu(x, y) + \mathfrak{m}(y)\mu(y, x)) = \mathfrak{m}(x)\mu(x, y),$$

where we abbreviate $\mathfrak{m}(x, y)$ as $\mathfrak{m}_{xy}$. Note that $\mathfrak{m}_{xy} = \mathfrak{m}_{yx}$ is symmetric and $\mu(x, y) = \frac{\mathfrak{m}_{xy}}{\mathfrak{m}(x)}$.

**Definition B.7.** For a non-empty $\Omega \subset \mathcal{V}$, its boundary measure is defined as

$$\mathfrak{m}(\partial \Omega) := \sum_{y \in \Omega} \sum_{z \in V \setminus \Omega} \mathfrak{m}_{yz},$$

where $\mathfrak{m}_{yz}$ is defined in B.6 and $\mathfrak{m}(\Omega) = \sum_{x \in \Omega} \mathfrak{m}(x)$. Then the *Dirichlet isoperimetric constant* $\mathcal{I}_{\mathcal{V}}^D$ on $\mathcal{V}$ is defined by

$$\mathcal{I}_{\mathcal{V}}^D := \inf_{\Omega} \frac{\mathfrak{m}(\partial \Omega)}{\mathfrak{m}(\Omega)},$$

which is analogous to cheeger constant on weighted-directed graph.

**Definition B.8.** Exploiting definition 4.3 on Continuous Unified Ricci Curvature , we introduce a limit version of **CURC** concerning idleness for theoretical derivation. Define the $\alpha$-*idle Mean Transition probability Kernel* by

$$\mu_x^{\alpha}(y) = \mu^{\alpha}(x, y) := \begin{cases} \frac{1}{2}\alpha[W(x, y) + \frac{\mathfrak{m}(y)}{\mathfrak{m}(x)}W(y, x)] & \text{if } y \neq x \\ (1 - \alpha) & \text{if } y = x \end{cases} \tag{9}$$

We use the same $\varepsilon$-*masked weighted edge length* for calulating Wasserstein distance. Then we define the $\alpha$-*idle $\varepsilon$-masked Continuous Unified Ricci Curvature* and $\alpha$-*idle Continuous Unified Ricci Curvature* by

$$\kappa_{\text{CURC}}^{\varepsilon\alpha}(x, y) := 1 - \frac{\mathcal{W}_1^{\varepsilon}(\mu_x^{\alpha}, \mu_y^{\alpha})}{d^{\varepsilon}(x, y)},$$

$$\kappa_{\text{CURC}}^{\alpha}(x, y) := \lim_{\varepsilon \to 0} \frac{\kappa_{\text{CURC}}^{\varepsilon\alpha}(x, y)}{\alpha}$$

Hence, we define the *idle-CURC* by

$$\kappa_{\text{CURC}}^I(x, y) := \lim_{\varepsilon \to 0} \lim_{\alpha \to 0} \frac{\kappa_{\text{CURC}}^{\varepsilon\alpha}(x, y)}{\alpha}.$$

**Theorem B.11.** *Let $\mathcal{G} = (\mathcal{V}, \mathcal{E}, \omega)$ be a strongly-connected weighted-directed graph, we have that for all $x \neq y \in \mathcal{V}$,*

$$\kappa_{\text{CURC}}^I(x, y) \geq \kappa_{\text{CURC}}(x, y).$$

*Proof.* Using the trick mentioned in the proof of B.10, we choose $\varepsilon$ sufficiently small such that $\kappa_{\text{CURC}}^{\varepsilon\alpha}(x, y)$ is independent of $\varepsilon$. WLOG, we abbreviate $\kappa_{\text{CURC}}^{\varepsilon\alpha}(x, y)$ as $\kappa_{\text{CURC}}^{\alpha}(x, y)$. By lemma 3.2 from (Ozawa et al., 2020), $\kappa_{\text{CURC}}^{\alpha}(x, y)$ is concave in $\alpha \in [0, 1]$ and $\frac{\kappa_{\text{CURC}}^{\alpha}(x,y)}{\alpha}$ is non-increasing in $\alpha \in (0, 1]$. Note that $\kappa_{\text{CURC}}(x, y)$ is nothing but $\frac{\kappa_{\text{CURC}}^{\alpha}(x,y)}{\alpha}$ taking $\alpha = 1$. Therefore by monotonicity, $\kappa_{\text{CURC}}^I(x, y) \geq \kappa_{\text{CURC}}(x, y)$. $\square$

**Proposition B.12.** *Let $\Omega \subset \mathcal{V}$ be a non-empty subset. Then for all function $f_0, f_1 : \mathcal{V} \to \mathbb{R}$,*

$$\sum_{x \in \Omega} \mathcal{L}f_0(x)f_1(x)\mathfrak{m}(x) = \frac{1}{2}\sum_{x,y \in \Omega}(f_0(y) - f_0(x))(f_1(y) - f_1(x))\mathfrak{m}_{xy}$$

$$- \sum_{x \in \Omega} \sum_{y \in V \setminus \Omega}(f_0(y) - f_0(x))f_1(x)\mathfrak{m}_{xy}.$$

*Proof.* The proof is a merely a calculation similar to integration by part. We stress that the result only depends on fact that $\Updownarrow$ is symmetric. (c.f. Theorem 2.1 in (Grigor'yan, 2018)) $\square$

**Lemma B.13.** *Let $x, y \in \mathcal{V}$ with $x \neq y$. Then*

$$\kappa^{\alpha}_{\mathrm{CURC}}(x,y) = \inf_{f \in \mathrm{Lip}_1(\mathcal{V})} \left( \frac{1}{\alpha} \left( 1 - \nabla_{xy} f \right) + \nabla_{xy} \mathcal{L} f \right),$$

*where $\nabla_{xy} f := \frac{f(y) - f(x)}{d(x,y)}$.*

*Proof.* We refer to lemma 3.9 in (Ozawa et al., 2020), which is essentially similar. It is worth noticing that we are using a different weighted distance function, but there is no restriction on the distance function in the proof. $\square$

**Proposition B.14.** *Let $x, y \in \mathcal{V}$ with $x \neq y$. Suppose $\mathcal{F}_{xy} := \{f \in \mathrm{Lip}_1(\mathcal{V}) \mid \nabla_{xy} f = 1\}$. Then we have*

$$\kappa^{I}_{\mathrm{CURC}}(x,y) = \inf_{f \in \mathcal{F}_{xy}} \nabla_{xy} \mathcal{L} f.$$

*Proof.* We refer to theorem 3.10 in (Ozawa et al., 2020), where the only part worth mentioning is that we require

$$\mathrm{Lip}_{1,x}(V) := \{f \in \mathrm{Lip}_1(V) \mid f(x) = 0\},$$

to be compact w.r.t. the canonical topology on $\mathbb{R}^n$. Let $\omega^* := \inf_{x,y \in \mathcal{V}} \omega(x,y) > 0$, then $d(x,y) \leq \frac{n}{\omega^*}$, which is bounded for fixed weight matrix $\omega$. As we restrict $f(x) = 0$, the 1-Lipshitz function $f$ w.r.t. the weighted shortest distance function $d$ is still bounded. Therefore, changing the distance function does not break compactness w.r.t. $\mathbb{R}^n$. $\square$

**Theorem B.15.** *Let $x \in V$. For $K \in \mathbb{R}$ we assume $\inf_{y \in V \setminus \{x\}} \kappa^{I}_{\mathrm{CURC}}(x,y) \geq K$. For $\Lambda \in (-\infty, 0)$ we further assume $\mathcal{H}_x \geq \Lambda$. Then on $V \setminus \{x\}$, we have*

$$\mathcal{L}\rho_x \geq K\rho_x + \Lambda$$

*Proof.* Fix a node $x \in \mathcal{V}$. Note that the distance function $\rho_x \in \mathcal{F}_{xy}$ defined in proposition B.14, as $\nabla_{xy}\rho_x = \frac{\rho_x(y) - \rho_x(x)}{d(x,y)} = 1$. Hence, for all $y \in \mathcal{V} \setminus \{x\}$,

$$K \leq \kappa^{I}_{CURC}(x,y) \leq \nabla_{xy}\mathcal{L}\rho_x = \frac{\mathcal{L}\rho_x(y) - \mathcal{L}\rho_x(x)}{d(x,y)} \leq \frac{\mathcal{L}\rho_x(y) - \Lambda}{d(x,y)}.$$

When $y = x$, the result is direct from $\mathcal{H}_x \geq \Lambda$. Hence, $\mathcal{L}\rho_x \geq K\rho_x + \Lambda$. $\square$

**Theorem B.16.** *Let $x \in V$. For $K \in \mathbb{R}$ we assume $\inf_{y \in V \setminus \{x\}} \kappa_{\mathrm{CURC}}(x,y) \geq K$. For $\Lambda \in (-\infty, 0)$ we also assume $\mathcal{H}_x \geq \Lambda$. For $D > 0$ we further assume $\mathrm{InRad}_x V \leq D$. Then for every $R > 0$ with $KR + \Lambda > 0$, we have*

$$\mathcal{I}^{D}_{E_R(x)} \geq \frac{KR + \Lambda}{D}$$

*Proof.* The proof is analogue to Proposition 9.6 in (Ozawa et al., 2020), while we include the proof utilizing previous results for completeness. By theorem B.11, $\kappa_{\mathrm{CURC}}(x,y) \geq K$ implies $\kappa^{I}_{\mathrm{CURC}}(x,y) \geq K$. Let $\Omega \subset E_R(x)$ be a non-empty vertex set. By Proposition B.12, we have

$$-\sum_{y \in \Omega} \mathcal{L}\rho_x(y)\mathfrak{m}(y) = \sum_{y \in \Omega} \sum_{z \in V \setminus \Omega} \left( \rho_x(z) - \rho_x(y) \right) \mathfrak{m}_{yz}$$

$$\geq -\sum_{y \in \Omega} \sum_{z \in V \setminus \Omega} \rho_x(y)\mathfrak{m}_{yz}$$

$$\geq -D\mathfrak{m}(\partial\Omega)$$

By theorem B.15, for all $y \in \Omega$,

$$\mathcal{L}\rho_x(y) \geq K\rho_x(y) + \Lambda \geq KR + \Lambda$$

Therefore,

$$\sum_{y \in \Omega} [\mathcal{L}\rho_x(y) - (KR + \Lambda)]\mathfrak{m}(y) \geq 0$$

$$\Rightarrow \sum_{y \in \Omega} \mathcal{L}\rho_x(y)\mathfrak{m}(y) \geq \sum_{y \in \Omega} KR\mathfrak{m}(y) = (KR + \Lambda)\mathfrak{m}(\Omega).$$

Note that we have $\sum_{y \in \Omega} \mathcal{L}\rho_x(y)\mathfrak{m}(y) \leq D\mathfrak{m}(\partial\Omega)$, which combined together yields

$$D\mathfrak{m}(\partial\Omega) \geq (KR + \Lambda)\mathfrak{m}(\Omega)$$

$$\Rightarrow \frac{\mathfrak{m}(\partial\Omega)}{\mathfrak{m}(\Omega)} \geq \frac{(KR + \Lambda)}{D}$$

$$\Rightarrow \mathcal{I}_{E_R(x)}^D \geq \frac{KR + \Lambda}{D}$$

$\square$

Before we end the discussion of properties of Continuous Unified Ricci Curvature, we point out that computing optimal-transportation based graph curvature can be computational intensive from solving a Linear programming problem similar to corollary B.3. In (Topping et al., 2022) they provide a lower bound as estimation for the Ollivier-ricci graph curvature. Likewise, we provide a universal lower bound for CURC using Kantorovich-Rubinstein duality and a more precise lower bound for CURC under stronger assumptions.

**Proposition B.17.** *Based on the construction of **CURC**, we choose $\varepsilon$ sufficiently small s.t. distance function $d^\varepsilon$ is independent of $\varepsilon$ and denote it as $d$. Let $\mathcal{D}(x, y) := \max\{d(x, y), d(y, x)\}$ be the largest weighted distance between nodes $x$ and $y$. For distinct node $x, y \in \mathcal{V}$, we have*

$$\kappa_{\mathrm{CURC}}(x, y) \geq -\frac{2\mathcal{D}(x, y)}{d(x, y)}(1 - \mu(x, y) - \mu(y, x))_+ + \frac{1}{d(x, y)}(d(x, y) + \mathcal{D}(x, y) - \mathcal{H}(y, x))$$

$$- \frac{\mathcal{D}(x, y) - d(y, x)}{d(x, y)}(\mu(x, y) + \mu(y, x)),$$

*where $\mathcal{H}(x, y)$ is defined by*

$$\widetilde{\mathcal{H}}(x) := -\sum_{y \in V} \mu(x, y)d(y, x)$$

$$\mathcal{H}(x) := -\sum_{y \in V} \mu(x, y)d(x, y)$$

$$\mathcal{H}(x, y) := -\sum_{y \in \mathcal{V}} \mu(x, y)d(x, y) - \sum_{y \in \mathcal{V}} \mu(x, y)d(y, x) = \widetilde{\mathcal{H}}(x) + \mathcal{H}(x).$$

*Proof.* The proof is a simple calculation of the Kantorovich-Rubinstein duality and we present the proof along the line of Proposition 6.1 in (Ozawa et al., 2020). By theorem B.2, we have

$$\mathcal{W}_1(\mu_x, \mu_y) = \sup_{f \in \mathrm{Lip}(1)} \sum_{z \in \mathcal{V}} f(x)(\mu_x(z) - \mu_y(z))$$

$$= \sup_{f \in \mathrm{Lip}(1)} \left\{ \left( \sum_{z \in V \setminus \{x\}} (f(z) - f(y))\mu(y, z) \right) - \left( \sum_{z \in V \setminus \{y\}} (f(z) - f(x))\mu(x, z) \right) \right.$$

$$\left. + (f(y) - f(x))(1 - \mu(x, y) - \mu(y, x)) \right\}.$$

For an arbitrary function $f \in \mathrm{Lip}(1)$ w.r.t. $d$, we have that

$$f(z) - f(y) \leq d(y, z), \quad f(z) - f(x) \geq -d(z, x), \quad |f(y) - f(x)| \leq \mathcal{D}(x, y),$$

Therefore,

$$
\begin{aligned}
\mathcal{W}_1\left(\mu_x, \mu_y\right) \leq & \sum_{z \in V \backslash\{x\}} d(y, z) \mu(y, z)+\sum_{z \in V \backslash\{y\}} d(z, x) \mu(x, z) \\
& +\mathcal{D}(x, y)|1-\mu(x, y)-\mu(y, x)| \\
= & (-\mathcal{H}_y-d(y, x) \mu(y, x))+\left(-\overleftarrow{\mathcal{H}}_x-d(y, x) \mu(x, y)\right) \\
& +\mathcal{D}(x, y)\left(2(1-\mu(x, y)-\mu(y, x))_{+}-(1-\mu(x, y)-\mu(y, x))\right) \\
= & \mathcal{H}(y, x)-d(y, x)(\mu(x, y)+\mu(y, x)) \\
& +\mathcal{D}(x, y)\left(2(1-\mu(x, y)-\mu(y, x))_{+}-(1-\mu(x, y)-\mu(y, x))\right) \\
= & 2 \mathcal{D}(x, y)(1-\mu(x, y)-\mu(y, x))_{+}-(\mathcal{D}(x, y)-\mathcal{H}(y, x)) \\
& +(\mathcal{D}(x, y)-d(y, x))(\mu(x, y)+\mu(y, x)) .
\end{aligned}
$$

Hence,

$$
\begin{aligned}
\kappa_{CURC}(x, y)= & 1-\frac{\mathcal{W}_1(\mu_x, \mu_y)}{d(x, y)} \\
\geq & 1-2 \frac{\mathcal{D}(x, y)}{d(x, y)}(1-\mu(x, y)-\mu(y, x))_{+}-\frac{1}{d(x, y)}(\mathcal{D}(x, y)-\mathcal{H}(y, x)) \\
& +\frac{(\mathcal{D}(x, y)-d(y, x))}{d(x, y)}(\mu(x, y)+\mu(y, x)) \\
= & -\frac{2 \mathcal{D}(x, y)}{d(x, y)}(1-\mu(x, y)-\mu(y, x))_{+}+\frac{1}{d(x, y)}(d(x, y)+\mathcal{D}(x, y)-\mathcal{H}(y, x)) \\
& -\frac{\mathcal{D}(x, y)-d(y, x)}{d(x, y)}(\mu(x, y)+\mu(y, x)) .
\end{aligned}
$$

$\square$

*Remark* B.18. To calculate this lower bound, after pre-processing distance function $d$ and the mean transition probability, the asymptotic complexity is $\mathcal{O}(n)$ for each node pair $x \neq y$. The pre-processing inclues a Floyd-Washall algorithm for shortest path which is $\mathcal{O}(n^3)$ and a power method for computing perron eigenvectors which is usually $\mathcal{O}(n^2)$. Therefore, the overall computational complexity is $\mathcal{O}(n^3)$

When strongly-connected $\mathcal{G}=(\mathcal{V}, \mathcal{E}, \omega)$ satisfies $(x, y) \in \mathcal{E} \iff (y, x) \in \mathcal{E}$, and we use edge length 1 for computing the shortest distance, **CURC** degenerates to the curvature definition in (Ozawa et al., 2020). Under this stronger assumption, we can achieve a tighter bound for $\kappa_{CURC}$ with a specific transportation plan utilizing the topology of local neighborhoods.

**Definition B.9.** For $x \sim y$, we define

- $\overrightarrow{\mathcal{N}}_x:=\{y \in V \mid x \rightarrow y\}, \overleftarrow{\mathcal{N}}_x:=\{y \in V \mid y \rightarrow x\}, \mathcal{N}_x:=\overrightarrow{\mathcal{N}}_x \cup \overleftarrow{\mathcal{N}}_x$, which are *inner neighborhood*, *outer neighborhood* and *neighborhood* respectively. If $\forall x, y \in \mathcal{V}, x \rightarrow y$ implies $y \rightarrow x$, then $\overrightarrow{\mathcal{N}}_x=\overleftarrow{\mathcal{N}}_x=\mathcal{N}_x$. Hence we use notation $\mathcal{N}_x$ to denote neighborhood for $x$.

- $\Delta(x, y):=\mathcal{N}_x \cap \mathcal{N}_y$ denotes set of common neighbors of node $x$ and $y$.

- $\square(x, y):=\{z \in \mathcal{N}_x \backslash \mathcal{N}_y, z \neq y: \exists w \in(\mathcal{N}_z \cap \mathcal{N}_y) \backslash \mathcal{N}_x\}$, which denotes the neighbors of $x$ forming 4-cycle based at $x \sim y$ without diagonals inside.

- $\square^{\mathrm{m}}(x, y):=\max \{|U|: U \subseteq \square(x, y), \exists \varphi: U \rightarrow \square(y, x), \varphi \in \mathcal{D}(U)\}$, and we use $\varphi^{\mathrm{m}}$ to denote one such optimal pairing between $\square(x, y)$ and $\square(y, x)$.

**Proposition B.19.** *On a strongly-connected weighted-directed locally graph $G=(\mathcal{V}, \mathcal{E}, \omega)$, where $\forall x, y \in \mathcal{V}$, if $x \to y \iff y \to x$, we have that if $x \sim y$, then*

$$
\kappa_{CURC}(x, y) \geq - \left( 1 - \mu(x,y) - \mu(y,x) - \sum_{z \in \Delta(x,y)} \mu(x,z) \vee \mu(y,z) - \sum_{z \in \square(x,y)} \mu(x,z) \wedge \mu(y, \varphi^{\mathrm{m}}(z)) \right)_+
$$

$$
- \left( 1 - \mu(x,y) - \mu(y,x) - \sum_{z \in \Delta(x,y)} \mu(x,z) \wedge \mu(y,z) - \sum_{z \in \square(x,y)} \mu(x,z) \wedge \mu(y, \varphi^{\mathrm{m}}(z)) \right)_+
$$

$$
+ \sum_{z \in \Delta(x,y)} \mu(x,z) \wedge \mu(y,z).
$$

*Remark* B.20. When $\mathcal{G} = (\mathcal{V}, \mathcal{E}, \omega)$ satisfies $x \to y \iff y \to x$, optimal transportation based curvature of $x \sim y$ is only dependent up to cycles of size at most 5. In theorem 6 of (Jost & Liu, 2014), they give a bound concerning the influence of triangles and in theorem 2 of (Topping et al., 2022), they extend the result concerning cycles of size 4. The proof of this lower bound is in line with these two theorems and we give a tighter lower bound for $\kappa_{CURC}$ concerning the influence of 4-cycles compared to proposition B.17 under this stronger assumption. The computational cost for computing this lower bound is at most $\mathcal{O}(n^4)$. We stress that under specific user case for **CURC** which requires lower computational cost, the lower bound estimations for $\kappa_{CURC}$ from proposition B.17 and proposition B.19 become handy.

# C  EXPRESSIVENESS OF GPNNS

## C.1  COLOR REFINEMENT(CR)

The 1-dimensional Weisfeiler-Lehman algorithm (1-WL), also referred to as the color-refinement algorithm, operates iteratively to determine a color mapping $\mathcal{X}_{\mathcal{G}} : \mathcal{V} \mapsto \mathcal{C}$ for a given graph $\mathcal{G} = (\mathcal{V}, \mathcal{E})$, where $\mathcal{C}$ represents the set of colors. Every vertex is initially assigned an identical color. During each ensuing iteration, a hash function is utilized by the 1-WL algorithm to amalgamate the current color of each vertex with the colors of its adjacent vertices, thereby updating the vertex's color. The algorithm persists in this process for a substantial number of iterations $T$, typically set to $T = |\mathcal{V}|$.

## C.2  PROTOTYPICAL GPNNS

It is well known that most of the MPNNs have an expressiveness upper bound of 1-WL (Xu et al., 2019; Morris et al., 2019a). For GPNNs, the situation is a bit different due to the fact that we have the propagation matrix depends on $f(\mathbf{A})$, which is any permutation equivariant function.

Here we set two prototypical GPNNs to analyze their expressiveness. First, we consider the GPNNs with homogeneous adjacency functions across all heads and layers and refer to them as **Homogeneous GPNNs**.

$$f = f^{h,l} \tag{10}$$

For generality, we also consider the case when multiple adjacency function $f$ is used in the model. An extra prototype model is defined with recurrence as **Layer-recurrent GPNNs**

$$\begin{aligned} f^l &= f^{h,l} \\ f^l &= f^{l+p} \end{aligned} \tag{11}$$

In order to reach the upper-bounded expressiveness, the model would be assumed with a sufficient number of heads and two **MLPs** with a sufficient layer and width for $\pi$ and $\phi$.

**Lemma C.1.** *(Xu et al., 2019), Lemma 5 If we assume that the set $\mathcal{X}$ is countable, a function $f : \mathcal{X} \mapsto \mathbb{R}^n$ can be established such that each bounded-size multiset $\hat{\mathcal{X}} \subset \mathcal{X}$ has a unique corresponding function $h(\hat{\mathcal{X}}) := \sum_{x \in \hat{\mathcal{X}}} f(x)$. Additionally, a decomposition of any multiset function g can be represented as $g(\hat{\mathcal{X}}) = \phi\left(\sum_{x \in \hat{\mathcal{X}}} f(x)\right)$ for some function $\phi$.*

**Proposition C.2.** *Homogeneous GPNNs Suppose GPNN model M has a fixed propagation function $f(\mathbf{A})$, with sufficient heads and layers, the expressiveness of M is upper-bounded by the iterative color-refinement*

$$\mathcal{X}_{\mathcal{G}}^{t+1}(v) = hash\{\{(\mathcal{X}_{\mathcal{G}}^t(u), f(A)_{vu}) : u \in \mathcal{V}\}\} \tag{12}$$

*Proof.* First, we prove that for any $f$ there exists a function $\pi_{base}$ which is invective from each entry of $f(A)$ to $\mathbb{R}$. Here we define the set of all possible values of

$$F_n := \{f(A)_{(v,u)} : A = \mathrm{adj}(\mathcal{G}), \mathcal{G} = (\mathcal{V}, \mathcal{E}), |\mathcal{V}| \leq n, (v, u) \in \mathcal{V}^2\} \tag{13}$$

For all graphs with no more than $n$ nodes, the total number of possible values of $f(A)_{vu} \in \mathbb{R}^d$ is finite and depends on $n$ and $f$, denoted as $|F_n| = N$. Given arbitrary bijection $\mathrm{id} : F_n \mapsto [N]$, By the Stone–Weierstrass Theorem applied to the algebra of continuous functions $C(\mathbb{R}^d, \mathbb{R})$ there exists a polynomial $\pi_{base}$ so that

$$\pi_{base}(\mathrm{f}) = \mathrm{id}(\mathrm{f}), \text{ for any } \mathrm{f} \in F_n \tag{14}$$

Now, we are ready to construct the $\pi^h$ using $\pi_{base}$ combined with the indicator function $\pi_1^h(d) := \mathbb{I}(d = h)$. For each head, we have $\pi^h = \pi_1^h \circ \pi_{base}$. By multiplying $\boldsymbol{P}$ with $\phi(\boldsymbol{X})$, we can recover the color-refinement of

$$\chi_{\mathcal{G}}^l(v) := \mathrm{hash}\left(\left(\chi_{\mathcal{G}}^{l;1}(v), \chi_{\mathcal{G}}^{l;2}(v), \cdots, \chi_{\mathcal{G}}^{l;|F_n|}(v)\right)\right),$$

$$\text{where } \chi_{\mathcal{G}}^{l,h}(v) := \left\{\left\{\chi_{\mathcal{G}}^{l-1}(u) : u \in \mathcal{V}, \mathrm{id}(f(A)_{(v,u)}) = h\right\}\right\}. \tag{15}$$

In detail: by applying **Lemma** C.1 to matrix multiplication, we fulfill the injective multiset function in 15.

$$X_v^{l,h} = \sum_{u \in \{\pi^h(f(A)_{v,u})=1\}} \phi(X_u^l) \tag{16}$$

By concatenating all the heads (injective) and passing them to an **MLP** update function, we can fulfill the hash function in 15. Before conclusion, we refer to the universal approximation theorem of **MLPs** (Hornik et al., 1989) to validate the use of **MLPs** to approximate constructed functions. Finally, it's easy to see that the color refinement in 15 is identical to 12 □

**Proposition C.3.** *Layer-recurrent GPNNs Suppose GPNN model M has a layer-dependent propagation function $f^l(\mathbf{A})$, repeats every p layer: $f^l = f^{l+p}$. With sufficient heads and layers, by stacking repetitions of such repetition, the expressiveness of M is upper-bounded by the iterative color refinement*

$$\mathcal{X}_{\mathcal{G}}^{t+1}(v) = hash\{\{(\mathcal{X}_{\mathcal{G}}^t(u), (f^1(A)_{vu}, f^2(A)_{vu}, \ldots, f^p(A)_{vu})) : u \in \mathcal{V}\}\} \tag{17}$$

In order to prove this, we would like to introduce two concepts for general CR algorithms, the stable colormap and the partition of a colormap.

For any CR algorithm, at each iteration, the color mapping $\chi_{\mathcal{G}}^t$ induces a partition of the vertex set $\mathcal{V}$ with an equivalence relation $\sim_{\chi_{\mathcal{G}}^t}$ defined to be $u \sim_{\chi_{\mathcal{G}}^t} v \Leftrightarrow \chi_{\mathcal{G}}^t(u) = \chi_{\mathcal{G}}^t(v)$ for $u, v \in V$. We call each equivalence class a color class with an associated color $c \in C$, denoted as $(\chi_{\mathcal{G}}^t)^{-1}(c) := \{v \in V : \chi_{\mathcal{G}}^t(v) = c\}$. Formally, we define the partition of any color mapping $\chi_{\mathcal{G}}$

**Definition C.1. (Partition)** The partition corresponding to $\chi_{\mathcal{G}}$ is the set $P(\chi_{\mathcal{G}}) = \{\chi_{\mathcal{G}}^{-1}(c) : c \in \mathcal{C}_{\mathcal{G}}\}$, where $\mathcal{C}_{\mathcal{G}} := \{\chi_{\mathcal{G}} : v \in V\}$. More specifically, if any element in $P(\chi_{\mathcal{G}}^1)$ is a subset of some element in $P(\chi_{\mathcal{G}}^2)$, we say that $P(\chi_{\mathcal{G}}^1)$ is at least as fine as $P(\chi_{\mathcal{G}}^2)$

It's easy to see due to the hash function, any color refinement iteration refines the partition $P(\chi_{\mathcal{G}}^t)$ to a finer partition $P(\chi_{\mathcal{G}}^{t+1})$. Since the number of vertices $|V| \leq n$, there must exist an iteration $T < |V|$ such that $P(\chi_{\mathcal{G}}^T) = P(\chi_{\mathcal{G}}^{T+1})$. Formally, we define the stable color mapping and stable partition

**Definition C.2. (Stable partition and stable color mapping)** Given a graph $\mathcal{G} = (\mathcal{V}, \mathcal{E})$, and an CR refinement $C \in \text{End}(\text{Hom}(\mathcal{V}, \mathcal{C}))$. Starting from $\chi_{\mathcal{G}}^0 = c_0$ (a constant initial color mapping), $\chi_{\mathcal{G}}^{t+1} = C(\chi_{\mathcal{G}}^t)$. There exist an iteration $T < |\mathcal{V}|$, such that $P(\chi_{\mathcal{G}}^T) = P(\chi_{\mathcal{G}}^{T+1})$. Such $P(\chi_{\mathcal{G}}^T)$ is called a stable partition denoted as $P_{\text{stable}}(C)$. Furthermore, we use $\chi_{\mathcal{G}}(C)$ to represent one of the many $\chi_{\mathcal{G}}^{T'}$ with $T' \geq T$, namely the stable color mapping.

CR algorithms decide if the graph pair $(\mathcal{G}, \mathcal{H})$ is isomorphic by comparing the color mapping $\chi_{\mathcal{G}}^T$ and $\chi_{\mathcal{H}}^T$. If the stable partition of CR iteration $C^1$ is finer than $C^2$ for any graphs with finite nodes, we can conclude that $C^1$ is more powerful than $C^2$. We refer to (Zhang et al., 2023b) for more detail.

*Proof.* Given **Proposition** C.2, we know that each GPNN layer $l \in [L]$ with $\hat{l} := l \mod p$ can (under sufficient layers and width conditions) fulfill the coloring process of

$$\mathcal{X}_{\mathcal{G}}^{l+1}(v) = \text{hash}\{\{(\mathcal{X}_{\mathcal{G}}^l(u), f^{\hat{l}}(A)_{vu}) : u \in \mathcal{V}\}\} \tag{18}$$

We simplify the notation of this color refinement iteration by $C^{\hat{l}} \in \text{End}(\text{Hom}(\mathcal{V}, \mathcal{C}))$

$$\mathcal{X}_{\mathcal{G}}^{l+1} = C^{\hat{l}}(\mathcal{X}_{\mathcal{G}}^l) \tag{19}$$

Now, we define the color refinement iteration for a full recurrent period $p$ for $l = kp, k \in \mathbb{N}$,

$$\mathcal{X}_{\mathcal{G}}^{l+p} = C^p \circ \cdots \circ C^1 \circ C^0(\mathcal{X}_{\mathcal{G}}^l) \tag{20}$$

Our goal is to show that the combined color refinement $C_{\text{comb}} = C^p \circ \cdots \circ C^1 \circ C^0$ is as powerful as the color refinement in 17 denoted as $C_{\text{concat}}$. In order to achieve that, we will compare the stable

partition $P(C_{\text{comb}})$ and $P(C_{\text{concat}})$ on an arbitrary graph $\mathcal{G} = (\mathcal{V}, \mathcal{E})$. For stable coloring $\chi_{\mathcal{G}}(C_{\text{comb}})$ and $\chi_{\mathcal{G}}(C_{\text{concat}})$. For $v_1, v_2 \in \mathcal{V}$, we will prove:

$$\chi_{\mathcal{G}}(C_{\text{comb}})(v_1) = \chi_{\mathcal{G}}(C_{\text{comb}})(v_2) \Leftrightarrow \chi_{\mathcal{G}}(C_{\text{concat}})(v_1) = \chi_{\mathcal{G}}(C_{\text{concat}})(v_2) \tag{21}$$

From the left to right, since the stable partition is unique and denoted as $P_{\mathcal{G}}(\chi_{\mathcal{G}}(C_{\text{comb}}))$. We have:

$$\chi_{\mathcal{G}}(C_{\text{comb}})(v_1) = \chi_{\mathcal{G}}(C_{\text{comb}})(v_2) \Leftrightarrow \exists S \in P_{\mathcal{G}}(\chi_{\mathcal{G}}(C_{\text{comb}})), \text{ s.t. } v_1, v_2 \in S \tag{22}$$

Thus we have $P_{\mathcal{G}}(\chi_{\mathcal{G}}(C_{\text{comb}})) = P_{\mathcal{G}}(C^1 \circ \chi_{\mathcal{G}}(C_{\text{comb}})) = \cdots = P_{\mathcal{G}}(C^p \circ \cdots \circ C^1 \circ \chi_{\mathcal{G}}(C_{\text{comb}}))$, with $v_1, v_2 \in S$ an element of each of the partitions. Which is equivalent to $C^i(\chi_{\mathcal{G}}(C_{\text{comb}}))(v_1) = C^i(\chi_{\mathcal{G}}(C_{\text{comb}}))(v_2)$, for $i \in [p]$. Write it with hash function notation we have $\text{hash}\{\{(\chi_{\mathcal{G}}(C_{\text{comb}})(u), f^i(\mathbf{A}_{\mathcal{G}})_{v_1 u}), u \in \mathcal{V}\}\} = \text{hash}\{\{(\chi_{\mathcal{G}}(C_{\text{comb}})(u), f^i(\mathbf{A}_{\mathcal{G}})_{v_2 u}), u \in \mathcal{V}\}\}$, for $i \in [p]$. Thus we can infer that

$$\text{hash}\{\{(\chi_{\mathcal{G}}(C_{\text{comb}})(u), (f^0(\mathbf{A}_{\mathcal{G}})_{v_1 u}, f^1(\mathbf{A}_{\mathcal{G}})_{v_1 u}, \ldots, f^p(\mathbf{A}_{\mathcal{G}})_{v_1 u})), u \in \mathcal{V}\}\}$$
$$= \text{hash}\{\{(\chi_{\mathcal{G}}(C_{\text{comb}})(u), (f^0(\mathbf{A}_{\mathcal{G}})_{v_2 u}, f^1(\mathbf{A}_{\mathcal{G}})_{v_2 u}, \ldots, f^p(\mathbf{A}_{\mathcal{G}})_{v_2 u})), u \in \mathcal{V}\}\}$$

Recall on the right-hand side, that the hash notation of $C_{\text{concat}}$ is

$$\text{hash}\{\{(\chi_{\mathcal{G}}(C_{\text{concat}})(u), (f^0(\mathbf{A}_{\mathcal{G}})_{v_1 u}, f^1(\mathbf{A}_{\mathcal{G}})_{v_1 u}, \ldots, f^p(\mathbf{A}_{\mathcal{G}})_{v_1 u})), u \in \mathcal{V}\}\}$$
$$= \text{hash}\{\{(\chi_{\mathcal{G}}(C_{\text{concat}})(u), (f^0(\mathbf{A}_{\mathcal{G}})_{v_2 u}, f^1(\mathbf{A}_{\mathcal{G}})_{v_2 u}, \ldots, f^p(\mathbf{A}_{\mathcal{G}})_{v_2 u})), u \in \mathcal{V}\}\}$$

Thus $P(C_{\text{comb}})$ is at least as fine as $P(C_{\text{concat}})$. By which we prove the $\chi_{\mathcal{G}}(C_{\text{comb}})(v_1) = \chi_{\mathcal{G}}(C_{\text{comb}})(v_2) \Rightarrow \chi_{\mathcal{G}}(C_{\text{concat}})(v_1) = \chi_{\mathcal{G}}(C_{\text{concat}})(v_2)$

It is straightforward from left to right since the hash notation of $C_{\text{concat}}$ implies each of the $C^i$ iterations holds. We have $\chi_{\mathcal{G}}(C_{\text{comb}})(v_1) = \chi_{\mathcal{G}}(C_{\text{comb}})(v_2) \Leftrightarrow \chi_{\mathcal{G}}(C_{\text{concat}})(v_1) = \chi_{\mathcal{G}}(C_{\text{concat}})(v_2)$. By proving 21, we conclude that $C_{\text{comb}}$ is at least as fine as $C_{\text{concat}}$ and vice versa, thus $C_{\text{comb}}$ is as powerful as $C_{\text{concat}}$.

$\square$

Expressiveness assessment without recurrence might be a standalone research topic thus beyond the discussion of this paper, leaving for future work.

# D  GENERALITY OF GPNN FRAMEWORK

## D.1  MPNNs AS SPECIAL CASES OF GPNNs

In this section, we demonstrate how to cast various existing MPNNs/GTs into our framework. For simplicity, we give examples with single-head architecture, which can be directly generalized to multi-head.

### D.1.1  GCN CASTED BY GPNN

Graph Convolution Networks (GCN) (Kipf & Welling, 2017) can be cast into the proposed framework.

$$
\begin{aligned}
&U(\rho(\mathbf{P}) \times \phi(\mathbf{X}), \mathbf{X}) := \rho(\mathbf{P}) \times \phi(\mathbf{X}) \\
&\pi(\mathbf{X}_{(u)}, \mathbf{X}_{(v)}, f(\mathbf{A})_{(u,v)}) := f(\mathbf{A})_{(u,v)} \\
&\rho(\mathbf{P}) := \operatorname{diag}(\mathbf{P1})^{-1/2}\,\mathbf{P}\,\operatorname{diag}(\mathbf{1}^{\mathsf{T}}\mathbf{P})^{-1/2} \\
&\phi(\mathbf{X}) := \mathbf{XW} \\
&f(\mathbf{A}) := \mathbf{A}
\end{aligned}
\tag{23}
$$

where $\operatorname{diag} : \mathbb{R}^n \to \mathbb{R}^{n^2}$ converts a vector to a diagonal matrix.

### D.1.2  GAT CAST BY GPNN

As an example, here we demonstrate how one can cast the Graph Attention Network (GAT) (Veličković et al., 2018) to the proposed framework.

$$
\begin{aligned}
&U(\rho(\mathbf{P}) \times \phi(\mathbf{X}), \mathbf{X}) := \rho(\mathbf{P}) \times \phi(\mathbf{X}) \\
&\pi(\mathbf{X}_{(u)}, \mathbf{X}_{(v)}, f(\mathbf{A})_{(u,v)}) = e^{\alpha(\mathbf{X}_{(u)}, \mathbf{X}_{(v)})} \cdot f(\mathbf{A})_{(u,v)} \\
&\rho(\mathbf{P}) = \operatorname{diag}(\mathbf{P1})^{-1}\,\mathbf{P} \\
&\phi(\mathbf{X}) = \mathbf{XW} \\
&f(\mathbf{A}) = \mathbb{1}_{\{1\}}(\mathbf{A})
\end{aligned}
\tag{24}
$$

where $\alpha : \mathbb{R}^d \times \mathbb{R}^d \to \mathbb{R}$ can be an arbitrary real-value function, including not limited to, linear-projection with output non-linearity (Veličković et al., 2018), scaled dot-product (Vaswani et al., 2017a), multi-layer perceptrons (Brody et al., 2022; Bahdanau et al., 2015) and more complicated ones (Brockschmidt, 2020; Monti et al., 2017).

### D.1.3  GATEDGCN

Similar to GAT, GatedGCN (Bresson & Laurent, 2018) can be cast into our framework.

$$
\begin{aligned}
&U(\rho(\mathbf{P}) \times \phi(\mathbf{X}), \mathbf{X}) := \rho(\mathbf{P}) \times \phi(\mathbf{X}) \\
&\pi(\mathbf{X}_{(u)}, \mathbf{X}_{(v)}, f(\mathbf{A})_{(u,v)}) = \operatorname{Sigmoid}\big(\alpha(\mathbf{X}_{(u)}, \mathbf{X}_{(v)})\big) \cdot f(\mathbf{A})_{(u,v)} \\
&\rho(\mathbf{P}) = \operatorname{diag}(\mathbf{P1})^{-1}\,\mathbf{P} \\
&\phi(\mathbf{X}) = \mathbf{XW} \\
&f(\mathbf{A}) = \mathbb{1}_{\{1\}}(\mathbf{A})
\end{aligned}
\tag{25}
$$

where $\alpha : \mathbb{R}^d \times \mathbb{R}^d \to \mathbb{R}$ can be an arbitrary real-value function.

## D.2  GRAPH REWIRING AS SPECIAL CASES OF GPNNs

### D.2.1  DREW

Drew introduces dynamic rewiring for MPNNs, leading to multiple propagation graphs in each layer, similar to multi-head architecture.

Raising GCN-variant as an example, the $K$th layer of Drew, can be interpreted as $K$-head propagation,

$$U([\rho(\mathbf{P}^k) \times \phi^k(\mathbf{X})]_{\text{concat}}^K, \mathbf{X}) := \sum_{k=1}^{K} \rho(\mathbf{P}^k) \times \phi^k(\mathbf{X})$$

$$\pi^k(\mathbf{X}_{(u)}, \mathbf{X}_{(v)}, f^k(\mathbf{A})_{(u,v)}) = f^k(\mathbf{A})$$

$$\rho(\mathbf{P}) = \mathbf{P} \tag{26}$$

$$\phi^k(\mathbf{X}) = \mathbf{X}\mathbf{W}^k$$

$$f^k(\mathbf{A}) = \mathbb{1}_{\{k\}}(d_{\mathcal{G}}(\mathbf{A}))(\text{diag}(\mathbf{A}\mathbf{1})^{-1/2} \mathbf{A} \, \text{diag}(\mathbf{1}^{\mathsf{T}}\mathbf{A})^{-1/2})$$

where $d_{\mathcal{G}} : \{0,1\}^{n^2} \to \mathbb{R}^{n^2}$ denotes an arbitrary graph distance function given $\mathbf{A}$. Drew utilizes the shortest-path distance; functions and matrices without subscripts denote that they are shared across heads.

### D.3 GRAPH TRANSFORMERS AS SPECIAL CASES OF GPNNS

Similar to GAT, Graph Transformers, e.g., Graphormer (Ying et al., 2021), SAN (Kreuzer et al., 2021b), GRIT (Ma et al., 2023) can be cast into our framework. The key difference is that Graph Transformers are not limited by first-hop neighborhood and introduce absolute/relative positional encoding to sense non adjacent nodes in aggregation,

In generalized propagation, one can recover the exact arbitrary Graph Transformers by

$$U(\rho(\mathbf{P}) \times \phi(\mathbf{X}), \mathbf{X}) := \rho(\mathbf{P}) \times \phi(\mathbf{X})$$

$$\pi(\mathbf{X}_{(u)}, \mathbf{X}_{(v)}, f(\mathbf{A})_{(u,v)}) = e^{\alpha(\mathbf{X}_{(u)}, \mathbf{X}_{(v)}, f(\mathbf{A})_{(u,v)})}$$

$$\rho(\mathbf{P}) = \text{diag}(\mathbf{P}\mathbf{1})^{-1} \mathbf{P} \tag{27}$$

$$\phi(\mathbf{X}) = \mathbf{X}\mathbf{W}$$

where $\alpha : \mathbb{R}^d \times \mathbb{R}^d \times \mathbb{R}^d \to \mathbb{R}$ can be arbitrary real-value function; $f(\mathbf{A})$ is referred to as the positional encoding function, such as RRWP (Ma et al., 2023), Shortest-path distance (Ying et al., 2021) and resistance distance (Zhang et al., 2023b).

We list the definition of $\alpha$ for two graph transformers as examples:

**Graphormer**

$$\alpha(\mathbf{X}_{(u)}, \mathbf{X}_{(v)}, f(\mathbf{A})_{(u,v)}) := (\mathbf{W}_1\mathbf{X}_{(u)})^{\mathsf{T}}(\mathbf{W}_2\mathbf{X}_{(v)}) + f(\mathbf{A})_{(u,v)} \tag{28}$$

where $f(\mathbf{A})_{(u,v)} : \{0,1\}^{n^2} \to \mathbb{R}$ is learnable scalers indexed by shortest-path distance between $u$ and $v$.

**GRIT**

$$a(\mathbf{X}_{(u)}, \mathbf{X}_{(v)}, f(\mathbf{A})_{(u,v)})) := \mathbf{wReLU}(\rho((\mathbf{W}_1\mathbf{X}_{(u)}+\mathbf{W}_2\mathbf{X}_{(v)})\odot\mathbf{W}_3 f(\mathbf{A})_{(u,v)}))+\mathbf{W}_4 f(\mathbf{A})_{(u,v)})) \tag{29}$$

where $f(\mathbf{A})_{(u,v)} : \{0,1\}^{n^2} \to \mathbb{R}^d$ denotes a positional encoding, RRWP; and $\rho(x) := \sqrt{\text{ReLU}(x)} - \sqrt{\text{ReLU}(-x)}$ is signed square-root.

### D.3.1 HYBRID TRANSFORMER

Hybrid Transformers, e.g., SAN (Kreuzer et al., 2021b) (local attention + non-local attention) and GraphGPS (Rampášek et al., 2022) (MPNN + global attention), can be simply cast to a multi-head version of GPNNs, one head as MPNNs and the other head as Graph Transformers, as aforementioned.

# E   EXPERIMENTAL DETAILS

## E.1   DETAILS OF GPNN-PE

### E.1.1   MODEL ARCHITECTURE

To verify our theoretical findings, we build up a purely structural-based GPNN, called GPNN-PE, based on the SOTA GTs - GRIT (Ma et al., 2023), by removing the *query-key architecture* which models the token similarity on node representations.

In each layer, we update node representations $\mathbf{x}_u \forall u \in \mathcal{V}$ and node-pair representations $\mathbf{e}_{u,v}, \forall u, v \in \mathcal{V}$. Similar to GRIT, we initialize these using the initial node features and our RRWP positional encodings: $\mathbf{x}_i = [\mathbf{x}_i' \| \mathbf{P}_{i,i}] \in \mathbb{R}^{d_h+K}$ and $\mathbf{e}_{i,j} = [\mathbf{e}_{i,j}' \| \mathbf{P}_{i,j}] \in \mathbb{R}^{d_e+K}$, where $\mathbf{x}_i' \in \mathbb{R}^{d_h}$ and $\mathbf{e}_{i,j}' \in \mathbb{R}^{d_e}$ are observed node and edge attributes, respectively; $\mathbf{P}_{i,j}$ is the relative positional encoding for graphs. Note that, if node/edge attributes are not present in the data, we can set $\mathbf{x}_i'/\mathbf{e}_{i,j}'$ as zero-vectors $\mathbf{0} \in \mathbb{R}^d$. We set $\mathbf{e}_{i,j}' = \mathbf{0}$ if there is no observed edge from $i$ to $j$ in the original graph.

We replace the original attention computation in GRIT with a multi-layer perceptron (MLP):

$$
\begin{aligned}
\hat{\mathbf{e}}_{i,j} &= \sigma(\mathbf{W}_1 \mathbf{e}_{i,j}) \in \mathbb{R}^{d'}, \\
\alpha_{ij} &= \text{Softmax}_{j \in \mathcal{V}}(\mathbf{W}_2 \hat{\mathbf{e}}_{i,j}) \in \mathbb{R}, \\
\hat{\mathbf{x}}_i &= \sum_{j \in \mathcal{V}} \alpha_{ij} \cdot (\mathbf{W}_3 \mathbf{x}_j + \mathbf{W}_4 \hat{\mathbf{e}}_{i,j}) \in \mathbb{R}^{d''},
\end{aligned}
\tag{30}
$$

where $\sigma$ is a non-linear activation (ReLU by default); $\mathbf{W}_1 \in \mathbb{R}^{d' \times d}$, $\mathbf{W}_2 \in \mathbb{R}^{1 \times d'}$, $\mathbf{W}_3 \in \mathbb{R}^{d'' \times d}$ and $\mathbf{W}_4 \in \mathbb{R}^{d'' \times d'}$ are learnable weight matrices.

Following GRIT, we retain the update of edges and the multiple heads (say, $N_h$ heads) without further specification:

$$
\begin{aligned}
\mathbf{x}_i^{\text{out}} &= \sum_{h=1}^{N_h} \mathbf{W}_O^h \hat{\mathbf{x}}_i^h \in \mathbb{R}^d, \\
\mathbf{e}_{ij}^{\text{out}} &= \sum_{h=1}^{N_h} \mathbf{W}_{\text{Eo}}^h \hat{\mathbf{e}}_{ij}^h \in \mathbb{R}^d,
\end{aligned}
\tag{31}
$$

where $\mathbf{W}_O^h, \mathbf{W}_{\text{Eo}}^h \in \mathbb{R}^{d \times d''}$ are output weight matrices for each head $h$.

### E.1.2   POSITIONAL ENCODING

In this work, we apply Relative Random walk positional encoding utilized in GRIT (Ma et al., 2023), which is one of the most expressive graph positional encoding. Let $\mathbf{A} \in \mathbb{R}^{n \times n}$ be the adjacency matrix of a graph $(\mathcal{V}, \mathcal{E})$ with $n$ nodes, and let $\mathbf{D}$ be the diagonal degree matrix. Define $\mathbf{M} := \mathbf{D}^{-1}\mathbf{A}$, and note that $\mathbf{M}_{ij}$ is the probability that $i$ hops to $j$ in one step of a simple random walk. The proposed relative random walk probabilities (RRWP) initial positional encoding is defined for each pair of nodes $i, j \in \mathcal{V}$ as follows:

$$
\mathbf{P}_{i,j} = [\mathbf{I}, \mathbf{M}, \mathbf{M}^2, \ldots, \mathbf{M}^{k-1}]_{i,j} \in \mathbb{R}^k,
\tag{32}
$$

in which $\mathbf{I}$ is the identity matrix. In other words, in GPNN-PE, $\psi : \mathbb{R}^{n \times n} \to \mathbb{R}^{n \times n \times k}$ is defined as

$$
\psi(\mathbf{A})_{i,j} := [\mathbf{I}, \mathbf{D}^{-1}\mathbf{A}, (\mathbf{D}^{-1}\mathbf{A})^2, \ldots, (\mathbf{D}^{-1}\mathbf{A})^{k-1}]_{i,j}
\tag{33}
$$

## E.2   DETAILS OF EXPERIMENTAL SETUP

### E.2.1   BASELINES

We primarily compare our methods with the SOTA graph transformer, GRIT (Ma et al., 2023), as well as a number of prevalent graph-learning models: popular message-passing neural networks (GCN (Kipf & Welling, 2017), GIN  (Xu et al., 2019) and its variant with edge-features (Hu

et al., 2020), GAT (Veličković et al., 2018), GatedGCN (Bresson & Laurent, 2018), GatedGCN-LSPE (Dwivedi et al., 2021), PNA (Corso et al., 2020)); Graph Transformers (Graphormer (Ying et al., 2021), K-Subgraph SAT (Chen et al., 2022), EGT (Hussain et al., 2022), SAN (Kreuzer et al., 2021b), Graphormer-URPE (Luo et al., 2022), Graphormer-GD (Zhang et al., 2023b), GraphGPS (Rampášek et al., 2022)); and other recent Graph Neural Networks with SOTA performance (DGN (Beani et al., 2021), GSN (Bouritsas et al., 2022b), CIN (Bodnar et al., 2021), CRaW1 (Toenshoff et al., 2021), GIN-AK+ (Zhao et al., 2021)).

### E.2.2    DESCRIPTIONS OF DATASETS

A summary of the statistics and characteristics of datasets is shown in Table. 4. The first five datasets are from Dwivedi et al. (2022a) and the last two are from Dwivedi et al. (2022b). Readers are referred to Dwivedi et al. (2022a) and Dwivedi et al. (2022b) for more details about the datasets.

Table 4: Overview of the graph learning datasets involved in this work (Dwivedi et al., 2022a;b; Irwin et al., 2012; Hu et al., 2021) .

| Dataset | # Graphs | Avg. # nodes | Avg. # edges | Directed | Prediction level | Prediction task | Metric |
|---------|----------|--------------|--------------|----------|------------------|-----------------|--------|
| ZINC | 12,000 | 23.2 | 24.9 | No | graph | regression | Mean Abs. Error |
| MNIST | 70,000 | 70.6 | 564.5 | Yes | graph | 10-class classif. | Accuracy |
| CIFAR10 | 60,000 | 117.6 | 941.1 | Yes | graph | 10-class classif. | Accuracy |
| PATTERN | 14,000 | 118.9 | 3,039.3 | No | inductive node | binary classif. | Weighted Accuracy |
| CLUSTER | 12,000 | 117.2 | 2,150.9 | No | inductive node | 6-class classif. | Accuracy |
| Peptides-func | 15,535 | 150.9 | 307.3 | No | graph | 10-task classif. | Avg. Precision |
| Peptides-struct | 15,535 | 150.9 | 307.3 | No | graph | 11-task regression | Mean Abs. Error |

### E.2.3    DATASET SPLITS AND RANDOM SEED

Our experiments are conducted on the standard train/validation/test splits of the evaluated benchmarks. For each dataset, we execute 4 runs with different random seeds (0,1,2,3) and report the mean performance and standard deviation.

### E.2.4    HYPERPARAMETERS

Due to the limited time and computational resources, we did not perform an exhaustive search or a grid search on the hyperparameters. We mainly follow the hyperparameter setting of GRIT (Ma et al., 2023) and make slight changes to fit the number of parameters into the commonly used parameter budgets. For the benchmarks from Dwivedi et al. (2022a;b), we follow the most commonly used parameter budgets: up to 500k parameters for ZINC, PATTERN, CLUSTER, Peptides-func and Peptides-struct; and ∼100k parameters for MNIST and CIFAR10.

The final hyperparameters are presented in Tables. 5 and Tables. 6.

### E.3    VISUALIZATION OF CURVATURE

In this section, we provide visualizations of attention maps and their corresponding CURC maps, as well as the distribution of CURC shown as the kernel density estimation (KDE) plots.

To better demonstrate the trend of the learned attention maps, we run a simplified variant of GPNN-PE, with single-head attention shared across layers, on ZINC (Dwivedi et al., 2022a), and visualize the first 6 graphs in the Test set as examples.

In Fig. 3,4,5,6,7,8, the 1st row is the visualization of the attention map; the 2nd row is the visualization of the CURC map; and the 3rd row is the KDE plot for the CURC.

We also visualize the trend curves of minimum/average CURC given the attention matrices across the first 32 test graphs in ZINC, as shown in Fig. 9 and Fig. 10.

Table 5: Hyperparameters for five datasets from BenchmarkingGNNs (Dwivedi et al., 2022a).

| Hyperparameter | ZINC | MNIST | CIFAR10 | PATTERN | CLUSTER |
|---|---|---|---|---|---|
| # Transformer Layers | 10 | 4 | 4 | 10 | 16 |
| Hidden dim | 64 | 52 | 52 | 64 | 48 |
| # Heads | 8 | 4 | 4 | 8 | 8 |
| Dropout | 0 | 0 | 0 | 0 | 0.01 |
| Attention dropout | 0.2 | 0.5 | 0.5 | 0.2 | 0.5 |
| Graph pooling | sum | mean | mean | − | − |
| PE dim (RW-steps) | 21 | 18 | 18 | 21 | 32 |
| PE encoder | linear | linear | linear | linear | linear |
| Batch size | 32/256 | 16 | 16 | 32 | 16 |
| Learning Rate | 0.001 | 0.001 | 0.001 | 0.0005 | 0.0005 |
| # Epochs | 2000 | 200 | 200 | 100 | 100 |
| # Warmup epochs | 50 | 5 | 5 | 5 | 5 |
| Weight decay | $1e-5$ | $1e-5$ | $1e-5$ | $1e-5$ | $1e-5$ |
| # Parameters | 417,877 | 100,754, | 98,238 | 353, 877 | 319,670 |
| # Param. (1 Head) | 385,237 | 108,866 | 106,350 | 389,717 | 351,926 |
| # Param. (Share Attn) | 311,381 | 92,330 | 89,814 | 315,861 | 283,670 |

Table 6: Hyperparameters for two datasets from the Long-range Graph Benchmark (Dwivedi et al., 2022b)

| Hyperparameter | Peptides-func | Peptides-struct |
|---|---|---|
| # Transformer Layers | 4 | 4 |
| Hidden dim | 96 | 96 |
| # Heads | 4 | 8 |
| Dropout | 0 | 0 |
| Attention dropout | 0.2 | 0.2 |
| Graph pooling | mean | mean |
| PE dim (walk-step) | 17 | 24 |
| PE encoder | linear | linear |
| Batch size | 32 | 32 |
| Learning Rate | 0.0003 | 0.0003 |
| # Epochs | 200 | 200 |
| # Warmup epochs | 5 | 5 |
| Weight decay | 0 | 0 |
| # Parameters | 332,142 | 338,315 |
| # Param. (1 Head) | 370,571 | 360,574 |
| # Param. (Share Attn) | 310,091 | 304,702 |

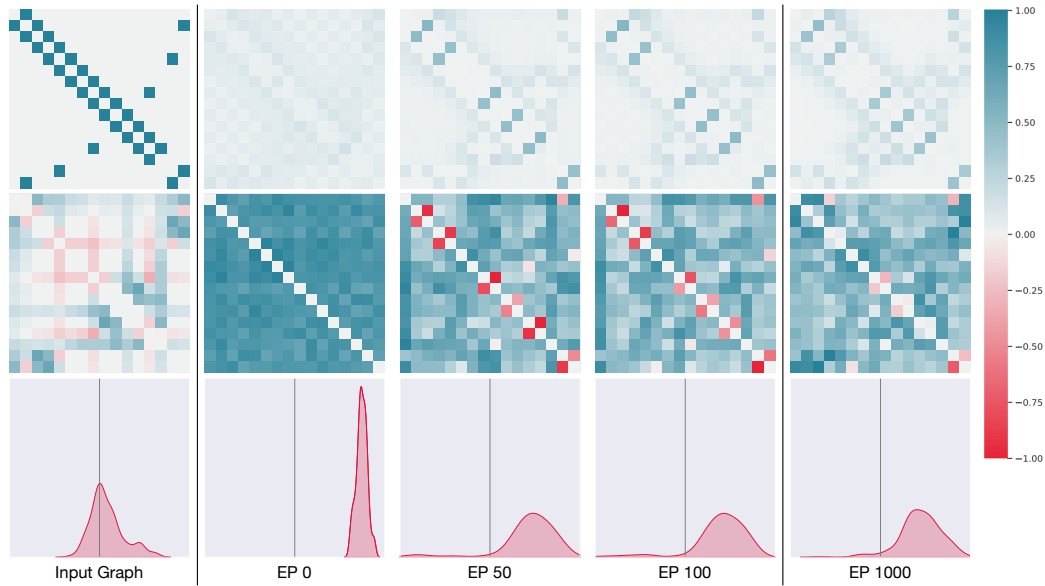

Figure 3: The visualization of Attention and CURC for Graph-1 in ZINC

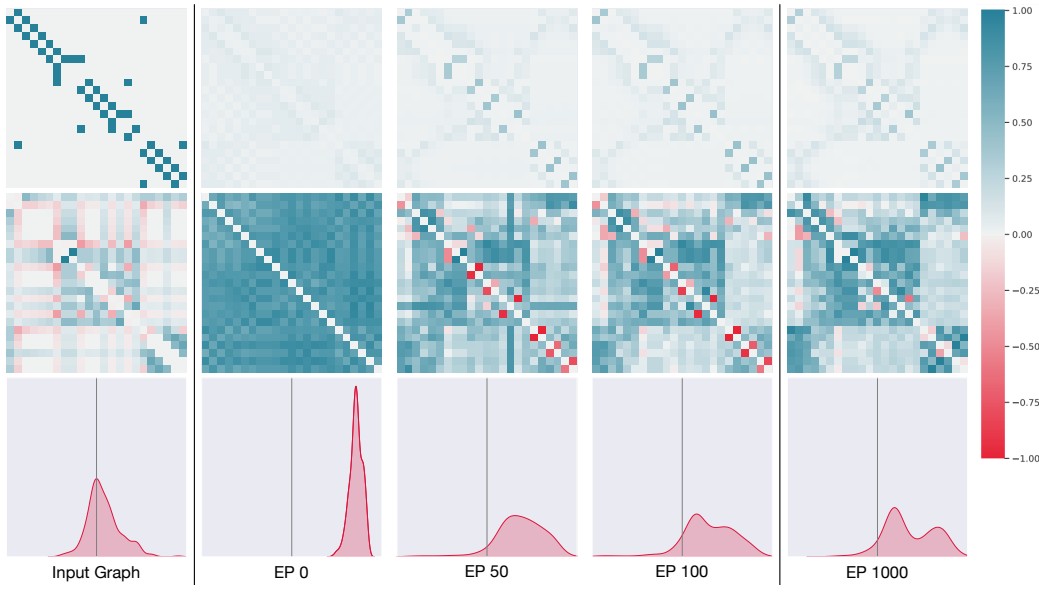

Figure 4: The visualization of Attention and CURC for Graph-2 in ZINC

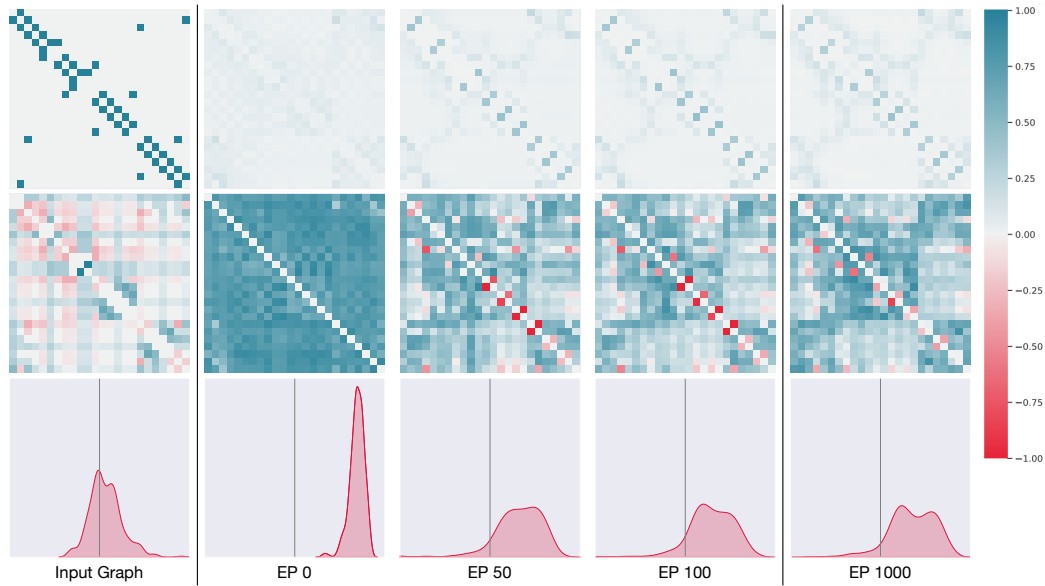

Figure 5: The visualization of Attention and CURC for Graph-3 in ZINC

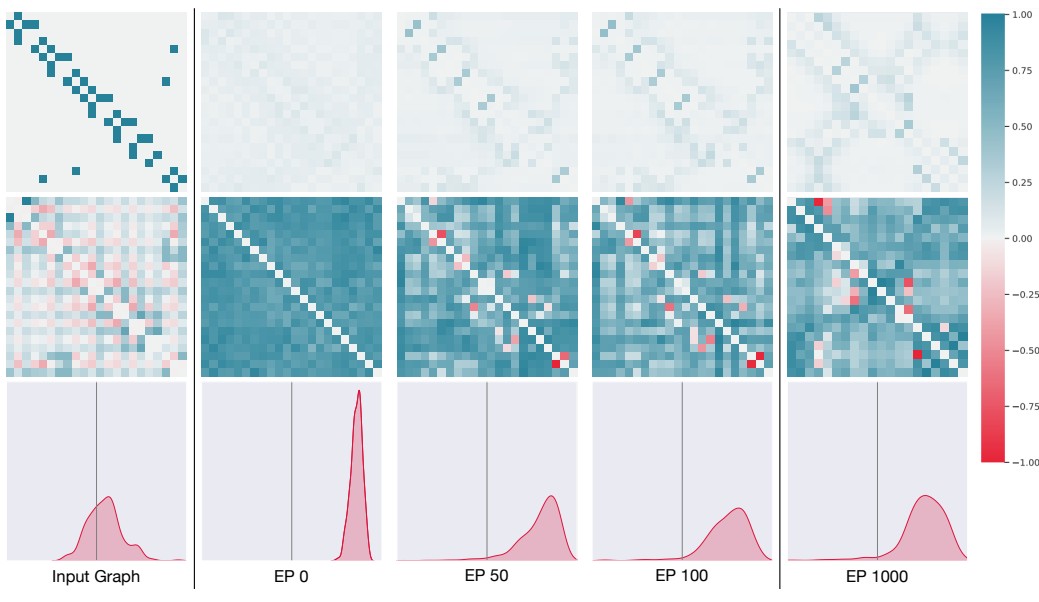

Figure 6: The visualization of Attention and CURC for Graph-4 in ZINC

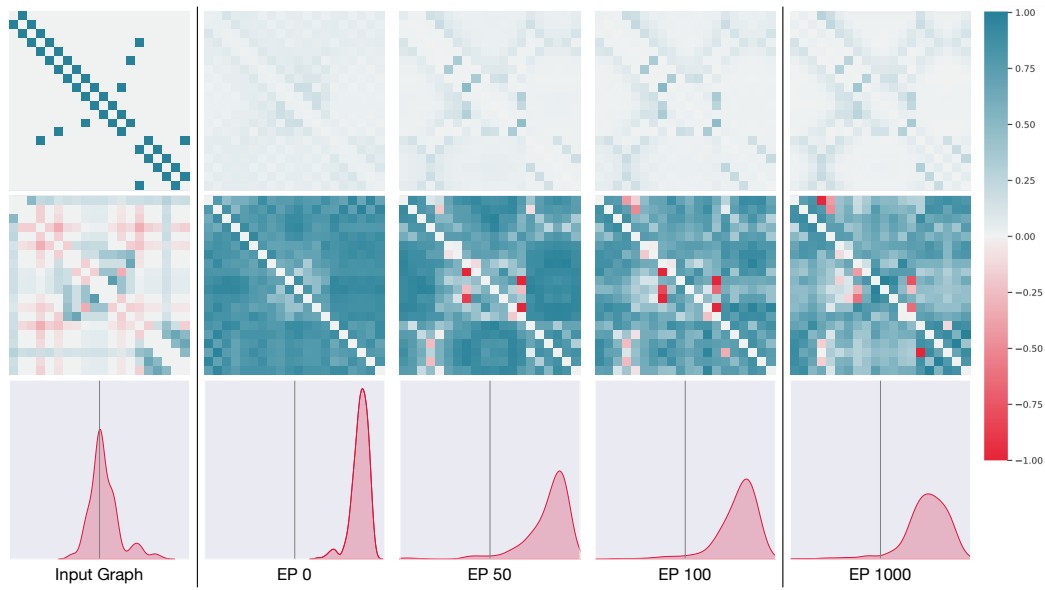

Figure 7: The visualization of Attention and CURC for Graph-5 in ZINC

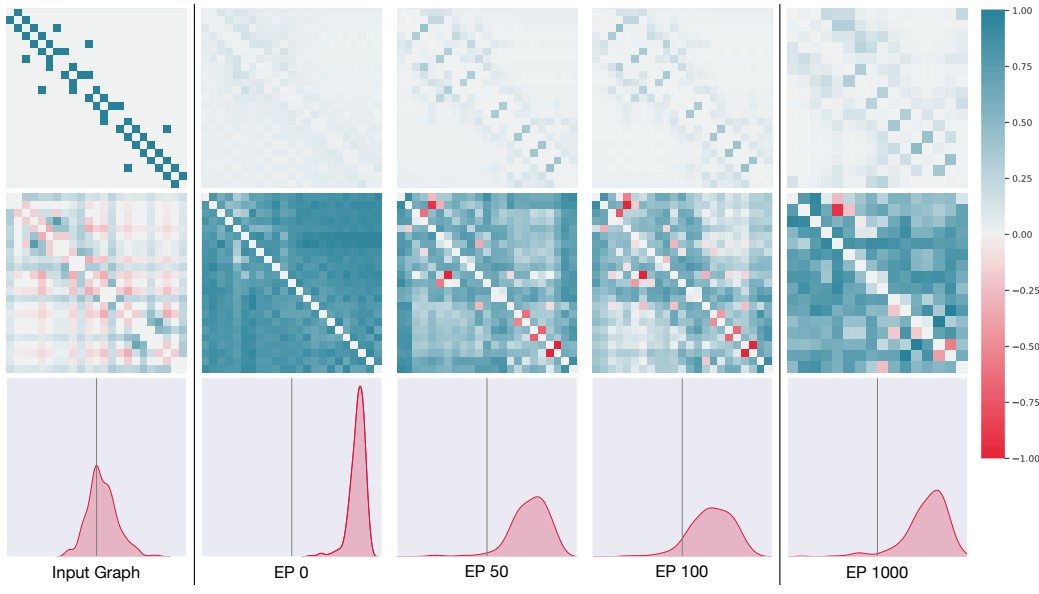

Figure 8: The visualization of Attention and CURC for Graph-6 in ZINC

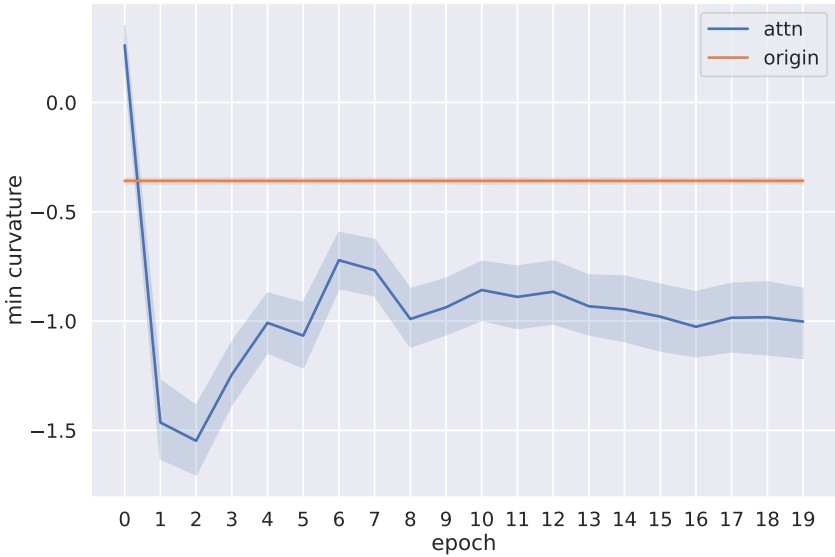

Figure 9: The trend of Minimum CURC for the first 32 test graphs in ZINC. Shade is the Confidence Interval at the 95% confidence level.

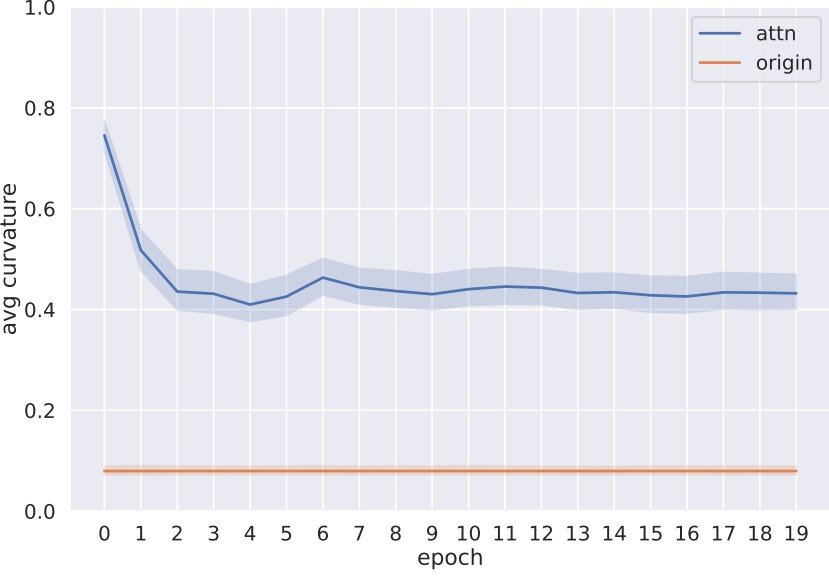

Figure 10: The trend of Average CURC for the first 32 test graphs in ZINC. Shade is the Confidence Interval at the 95% confidence level.

