# OpenReview forum: "Understanding Graph Transformers by Generalized Propagation"
_ICLR.cc/2024/Conference — Submitted to ICLR 2024_

### Official Review · Reviewer_6YEa · 2023-10-26

**Soundness:** 2 fair
**Presentation:** 2 fair
**Contribution:** 2 fair
**Rating:** 3
**Confidence:** 2

**Summary:**

This work proposes GPNNs, a framework unifying graph transformers, graph rewiring, and MPNNs. It shows the expressivity of GPMM equals to a color-refinement algorithm with adjacency function. Furthermore, it proposes a continuous extension of the Ollivier-Ricc curvature for analyzing the information propagation.

**Strengths:**

1. A unfied framework for MPNN, graph rewiring, and graph transformer.

**Weaknesses:**

1. The equivalence between Graph Transformer and the color refinement algorithm in proposition 3.1 are proved in previous work [1].

2. Section 3.3 looks incomplete. At end it mentions various GNN expressivity hierarchy without clarifying GPNN's connection with them.

3. Figures are not mentioned in the maintext.

4. The connection between CURC and GPNN seems fully intuitive. More formal connection can make me understand better.

5. The definition and properties of CURC seem quite straighforward. However, therefore meaning for GNN are not quite clear to me. How to use these theories to guide the design of GNN?

[1] Wenhao Zhu, Tianyu Wen, Guojie Song, Liang Wang, Bo Zheng, On Structural Expressive Power of Graph Transformers. KDD 2023. (released on Arxiv in May)

**Questions:**

1. The formal connection between CURC and GPNN.

2. More clear and detailed implication of CURC properties.

---

> ### Author Response · Authors · 2023-11-22
>
> Firstly, the authors would like to thank the reviewers for their detailed comments. GPNNs and CURC are both brand new concepts introduced by this work, thus the authors thank the reviewer for acknowledging their extensiveness. The authors will revise their paper to clarify the presentation accordingly. For the pointed out "weakness" and question, they would like to address as follows:

---

> ### Author Response · Authors · 2023-11-22
>
> > (W1) The equivalence between Graph Transformer and the color refinement algorithm in proposition 3.1 are proved in previous work [1].
>
> A: Firstly, we would argue that [1] shall be considered as a concurrent work since [1] was officially public in Aug 2023 and our submission was in Sep 2023. It is nearly impossible to track all new works on arXiv because of the tremendous amount of papers posted on arXiv.
>
> Second, the goal of our work is different from [1]. [1] aims to demonstrate that the proposed positional encoding together with global self-attention is endowed with stronger expressiveness compared to the 1-WL algorithm.
>
> In contrast, our work aims to illustrate that the adjacency function of various GNNs (e.g., MPNNs, graph transformers, etc.) is the dominant component in the expressiveness of modeling graph-structure, demonstrated with color refinement algorithms, while the feature-conditioning counterpart in self-attention (i.e., attention scores based on features similarity) is not the key factor.

---

> ### Author Response · Authors · 2023-11-22
>
> > (W2) Section 3.3 looks incomplete. At end it mentions various GNN expressivity hierarchy without clarifying GPNN's connection with them.
>
> A: Thank you for pointing out this. We would add more bridging text mentioning the connection to various GNNs. Besides, we did indeed include several examples to show various typical GNNs (including but limited to GCNs, GATs, and graph transformers), as special cases of GPNNs. The proposed GPNN framework allows us to analyze their expressiveness by simply analyzing the adjacency function.

---

> ### Author Response · Authors · 2023-11-22
>
> > (W3) Figures are not mentioned in the maintext.
>
> A: We would add bridging text for them.
>
> ----------------------
> &nbsp;
>
> > (W4) The connection between CURC and GPNN seems fully intuitive. More formal connection can make me understand better.
>
> A: Understanding the "bottleneck" of GNNs has been extensively studied and is known to be a critical factor affecting the performance of graph models [2]. In such models, bottlenecking refers to a situation where a small number of edges or vertices significantly restrict the flow or connectivity of information propagated by GNNs between two nodes.
> The authors' proposal in [2] to use Ollivier-Ricci (OR) curvature as a means to quantify this phenomenon has been a significant advancement in understanding graph structures.
> Due to the limitation of OR, they utilize OR on the input graphs which are unweighted and undirected to study the bottleneck of MPNNs.
> However, when the scope goes broader to various GNNs (e.g., MPNNs, graph rewiring, graph transformers), it is insufficient. Different GNNs behave differently given the same input graphs.
> Therefore, we propose a unified framework, GPNN, for various GNNs. Theoretically and empirically, we demonstrate that various GNNs can be viewed as generalized propagations with learned or fixed propagation matrices. Thus, we can study the bottleneck on GNNs directly via the corresponding propagation matrices instead of on the input graphs.
> However, the propagation graphs are naturally weighted-directed (for example, the learned attention matrices), where the OR is not applicable.
> This drives us to extend the OR to weighted-directed graphs, termed the CURC (Curvature Under Ricci Curvature), to become a valuable tool for studying the bottleneck of graph transformers.

---

> ### Author Response · Authors · 2023-11-22
>
> > (W5) The definition and properties of CURC seem quite straighforward. However, therefore meaning for GNN are not quite clear to me. How to use these theories to guide the design of GNN ?
>
> A: We would strongly argue that the definition and properties of CURC are not trivial. Even though curvatures have been utilized to analyze the bottleneck of MPNNs (Topping et al, ICML 2022). However, it is limited on analyzing the input graphs, which need to be simple graphs (i.e., undirected without self-loop) due to the property of curvature (only applicable the symmetric graphs). Considering the behaviors of various GNNs (e.g., MPNNs, graph rewiring, graph transformers, etc.) are not strictly tied to the input graphs: MPNNs and graph transformers, as examples, show different behaviors given the same input graphs.
> To better analyze various GNNs, we propose the unified framework, GPNN, which accommodates various GNNs. Furthermore, we demonstrate that the dominant component in GPNN to the expressiveness is the adjacency functions, which allow us to analyze the expressiveness of various GNNs in a unified approach, ridding of the less relevant parts.
> However, the propagation graphs from adjacency functions are naturally weighted and directed (e.g., attention matrices are asymmetric). To better understand the bottleneck of various GNNs under the GPNN framework, we carefully extend the definition of Ricci-curvature to weighted directed graphs without breaking the desired properties of curvature.
>
> The proposed tools, CURC and GPNN framework, are beneficial to the graph research community, for better understanding the behaviors of GNNs and guiding the future design of GNNs. Open questions can be studied with CURC and GPNN including but limited to: 1. whether the larger average curvatures of the propagation graphs are beneficial for GNNs to learn graphs? 2. do negative curvatures for node pairs contain important information on modeling graph structure?

---

> ### Author Response · Authors · 2023-11-22
>
> ----------------
> > (Q1) The formal connection between CURC and GPNN.
>
> A: Please find the answer for W5.
>
> > (Q2) More clear and detailed implication of CURC properties.
> A: Thank you for pointing out that the discussion on CURC could be more comprehensive in the main paper. In response, we are committed to refining the terminology and clarifying the indications for reader-friendliness. Additionally, we encourage readers to refer to Appendix B, where CURC is explained in detail. This should provide a more thorough understanding of the topic.
>
>
>
> **Again, thank you for your insightful and comprehensive advice that greatly improved our work. We will revise the apper to carry out the suggested experiments and make every effort to address the concerns and weaknesses. If there are any further questions or concerns, please let us know. If you feel that we have successfully addressed your concerns, please kindly consider a re-evaluation. Thank you.**

---

### Official Review · Reviewer_Wt6r · 2023-10-31

**Soundness:** 2 fair
**Presentation:** 2 fair
**Contribution:** 3 good
**Rating:** 6
**Confidence:** 4

**Summary:**

This paper introduces a unified view by defining the generalized propagation graph which is a weighted directed graph constructed from the input graph. By configuring the adjacency function f and entry-wise function \pi, various GNNs (MPNNs, graph rewiring and GTs) can be unified into a general framework, generalized propagation networks (GPNNs). And the authors show that the expressiveness of models within GPNN framework sorely depends on the adjacency function f. Therefore, novel GNN models with diverse expressive power can be constructed by designing appropriate adjacency functions. Extensive experiments are conducted on several public datasets to verify the effectiveness of the proposed model.

**Strengths:**

1.	The idea of this paper is interesting and clear. .
2.	The proposed method seems sensible, and perform well on several benchmarks.
3.	The paper provides a lot of theoretical analysis to support their claims.

**Weaknesses:**

1.	Generally, the paper is not very friendly to readers and need to be polished, especially section 3 and section 4.
2.	Deeper analysis about the experimental results in Table 2 and Table 3 are missed.
3.	More experiments need to be conducted on larger graph benchmarks (cora, Citeseer, Pubmed and OGB datasets) based on different GTs (except GRIT) to validate the effectiveness and scalability of the proposed method on large graphs.
4.	The paper is not very easy to follow and some notations are confusing. And some formulas are not numbered. Specifically, the format of Table 3 seems need to be reorganized.

**Questions:**

Please refer to the weaknesses part.

---

> ### Author Response · Authors · 2023-11-22
>
> Firstly, the authors would like to thank the reviewers for their detailed comments. GPNNs and CURC are both brand new concepts introduced by this work, thus the authors thank the reviewer for acknowledging their extensiveness. The authors will revise their paper to clarify the presentation accordingly. For the pointed out "weakness" and question, they would like to address as follows:

---

> ### Author Response · Authors · 2023-11-22
>
> > (W1) Generally, the paper is not very friendly to readers and needs to be polished, especially section 3 and section 4.
>
> A: We are currently working on improving the reader-friendliness by introducing clearer notation explanations and adding more bridging language for the revised version.

---

> ### Author Response · Authors · 2023-11-22
>
> > (W2) Deeper analysis about the experimental results in Table 2 and Table 3 are missed.
>
> A: We will add further discussion regarding the findings from Table 3. In general, the goal of Table 2 and Table 3 is to justify our hypothesis that the adjacency function is the essential component of the expressiveness of GNNs,  while the _feature conditioning_ property in self-attention, which is considered crucial in other domains, is just the icing on the cake with respect to the expressiveness in graph transformers. The hypothesis is verified by the observation that GPNN-PE without feature conditioning outperforms previous graph transformers with different adjacency functions and only slightly underperforms the graph transformer with the same adjacency function, GRIT.

---

> ### Author Response · Authors · 2023-11-22
>
> > (W3) More experiments need to be conducted on larger graph benchmarks (cora, Citeseer, Pubmed and OGB datasets) based on different GTs (except GRIT) to validate the effectiveness and scalability of the proposed method on large graphs.
>
> A: We would argue that scalability is actually not the focus of this work; this work aims to analyze and understand why graph transformers are efficacious on various graph learning tasks and which components in the architecture contribute to the efficacy. Therefore, we mainly focus on benchmarks widely utilized in evaluating the expressiveness of graph transformers in previous works (Kreuzer et al., NeurIPS 2021; Rampasek et al., NeurIPS 2022; Ma et al., ICML 2023).
> Thus, we would argue that addressing the scalability limitation of graph transformers is not the duty of this paper.
>
> In addition, the suggested node-level benchmarks (e.g., Cora, Citeseer, and Pubmed) are generally not suitable for evaluating the expressiveness of GNNs: even simple GNNs like GCNs, GATs are prone to overfitting, which obfuscates the understanding of expressiveness with the overfitting issue due to the limited amount of training data.

---

> ### Author Response · Authors · 2023-11-22
>
> > (W4) The paper is not very easy to follow and some notations are confusing. And some formulas are not numbered. Specifically, the format of Table 3 seems need to be reorganized.
>
> A: Same as the reply for W1. We would revise the unclear notations and number all formulas. Regarding the format of Table 3, we would greatly appreciate further suggestions on modification from the reviewers. We just simply followed the format utilized in previous works (Rampasek et al., NeurIPS 2022; Ma et al., ICML 2023).
>
>
>
> &nbsp;
>
> --------------
>
>
>
> **Again, thank you for your insightful and comprehensive advice that greatly improved our work. We will revise the apper to carry out the suggested experiments and make every effort to address the concerns and weaknesses. If there are any further questions or concerns, please let us know. If you feel that we have successfully addressed your concerns, please kindly consider a re-evaluation. Thank you.**
>
>
> - Kreuzer, D., Beaini, D., Hamilton, W., Létourneau, V., & Tossou, P. (2021). Rethinking graph transformers with spectral attention. Advances in Neural Information Processing Systems, 34, 21618-21629.
>
> - Rampášek, L., Galkin, M., Dwivedi, V. P., Luu, A. T., Wolf, G., & Beaini, D. (2022). Recipe for a general, powerful, scalable graph transformer. Advances in Neural Information Processing Systems, 35, 14501-14515.
>
> - Ma, L., Lin, C., Lim, D., Romero-Soriano, A., Dokania, P.K., Coates, M., Torr, P. &amp; Lim, S.. (2023). Graph Inductive Biases in Transformers without Message Passing. <i>Proceedings of the 40th International Conference on Machine Learning</i>, in <i>Proceedings of Machine Learning Research</i> 202:23321-23337

---

### Official Review · Reviewer_E8vF · 2023-11-01

**Soundness:** 3 good
**Presentation:** 3 good
**Contribution:** 3 good
**Rating:** 5
**Confidence:** 4

**Summary:**

In this paper, the authors introduce a general framework for representing Graph Neural Networks (GNNs), Message Passing Neural Network (MPNNs), Graph Transformers (GTs) and graph rewirings. They call it Generalized Propagation Neural Network (GPNN) and it can be specialized/instantiated by defining a function of the adjacency matrix of the graph to learn and an entry-wise function for pairs of graph nodes. They show that expressiveness of the model depends only on the function of its adjacency and they empirically test this by removing self-attention (i.e. the entry-wise function) from one of the GT models  and still retaining its performance (prior to its modification). The authors also extend Ollivier-Ricci (OR) curvature to weighted directed graphs, thus defining Continuous Unified Ricci Curvature (CURC). They study the theoretical properties of CURC and leverage this in analyzing the shifts in curvature distribution for propagation graphs after training, for a number of graph learning models that can be cast as GPNNs.

**Strengths:**

- This is a rich/extensive and ambitious work (both in theory and experiments - particularly in theoretical developments), around two key notions: GPNN and CURC. These notions are novel and serve as interesting additions to the expanding neural graph learning literature.

- Structure and high-level flow is smooth; content is reasonably split across the main manuscript text and its appendix. This is particularly useful in cases like this when a broad set of definitions and theorems must be combined.

**Weaknesses:**

- The presentation of GPNN can be considerably simplified/clarified. In Section 3.2 in particular: (a) using fewer symbols (are both $P$ and $\pi$ absolutely necessary?), (b) providing standard names to symbols so that referring to them is straightforward (e.g. $\rho()$ is referred to as both "normalized function" and "normalized propagation" which may be confusing, $\phi()$ does not have a descriptive name), (c) consider using a simple model as GNN as an example of the choices for the functions "embedded" within the text (rather than only as part of Table 1).

- CURC could be illustrated (and  ontrasted to OR) using a very small, simple example graph. The reader would then get an intuitive understanding of why CURC is strictly necessary in this context and also be able to better understand its algebraic and geometric properties (Section 4.3)

- It is not clear how the connection of the expressiveness of the model to (only) the function of its adjacency matrix could drive the choice of particular forms for such appropriate functions (which would certainly be a highly practical implication of this work).

**Questions:**

- Challenging the benefits of global self-attention in graph transformers (GTs) is a very strong statement. How could this reconcile with the reported elevated efficacy of GTs (i.e. with global self-attention) in various graph learning tasks in the literature (relative to GNNs, which do not have global self-attention)? The reader would be interested to know of any other potential empirical factors that could account for this conclusion.

Minor typos
- Page 3: entries-wise -> entry wise
- Page 4: identical to 2 -> identical to Equation (2)
- Page 4: every p layer -> every p layers
- Page 9: optical CURC distribution -> ?(optical)

---

> ### Author Response · Authors · 2023-11-22
>
> Firstly, the authors would like to thank the reviewers for their detailed comments. GPNNs and CURC are both brand new concepts introduced by this work, thus the authors thank the reviewer for acknowledging their extensiveness. The authors will revise their paper to clarify the presentation accordingly. For the pointed out weakness and question, they would like to address as follows:

---

> ### Author Response · Authors · 2023-11-22
>
> > (W1 a) The presentation of GPNN can be considerably simplified/clarified ... are both $P$ and $\pi$ absolutely necessary ...
>
> A: The reason for keeping the matrix-form output $P$ with the entry-wise function $\pi$ is two-fold. Firstly, we would like to maintain matrix notation to shorten the expression of $\rho$ $f$ and $\phi$ in eq. 23-29 (appendix D). With a matrix notation, those expressions can be significantly shorter than their entry-wise counterparts. Secondly, the GPNN framework started from the idea of separating the entry-wise function $\pi$ with adjacent function $f$. Thus, we would like to embed the restriction of "entry-wise" in the notation of $\pi$ to make sure the theory would hold with various GNNs that are able to be cast as GPNNs. Thus, from our point of view, it is unavoidable to use an entry-wise notation for $\pi$ in eq. 2, but keep matrix notation for the others. $P$ matrix and $P_{u,v}$ entry is the variable that connects the entry-wise notion and matrix notion.
>
>
> > (W1 b) providing standard names to symbols so that referring to them is straightforward (e.g. $\rho$() is referred to as both "normalized function" and "normalized propagation" which may be confusing,  does not have a descriptive name)
>
> A: We thank the reviewer for the very detailed example of the name ambiguity that appears. From our inspection, we found the specific case the reviewer mentioned is in Sec 3.2:
>
> **"normalized function"** -- "And $\rho$, which represents the normalized function."
> **"normalized propagation"** -- "where $\times$ denotes the matrix multiplication between normalized propagation $\rho(P)$ and $\phi(X)$".
>
> As stated in the former sentence, $\rho$ is referred to as a normalized function. In the second, we are referring to the matrix output of $\rho(P)$ as normalized propagation. Thus, there is actually no two-name issue associated with the same function. We will include extra clarification to reduce the ambiguity it might cause.
>
> > (W1 c) Consider using a simple model as GNN as an example of the choices for the functions "embedded" within the text (rather than only as part of Table 1)
>
> A: We would like to direct the reviewer to Appendix D, where we demonstrate how to cast various existing MPNNs/GTs/Rewirings into our framework. The model which we included is various:
> **MPNNs** -- GCN, GAT, GatedGCN
> **GTs** -- Graphormer, GRIT, SAN, GraphGPS
> **Rewiring** -- DREW
> We have included an extra paragraph in Sec 3.2 to emphasize the importance of those examples and encourage the readers towards Appendix D.

---

> ### Author Response · Authors · 2023-11-22
>
> > (W2) CURC could be illustrated using a very small, simple example graph ... get an intuitive understanding ...  its algebraic and geometric properties.
>
> A: We agree with the reviewer that the visualization of CURC could be improved. Illustrating CURC on a **small simple example graph** is possible, however, picking the specific graph would highly depend on what properties of CURC one would like to show. However, contrasting **OR** could be difficult since OR can only be applied on unweighted-undirected graphs while CURC is extended from OR to cover weighted digraphs. Thus CURC is equivalent to OR where OR is applicable. Given those difficulties, in the paper, we took another approach in **Figure 1**, where we demonstrate **CURC-curvature** (with top-10/20/30% Attention) on **a ZINC graph** with practical model (GPNN-PE). Such visualization concentrates more on the exact application of CURC is considered in this work: to analyze the information flow in a learned propagation graph. We will include a simple example graph in the appendix to show CURC's properties.

---

> ### Author Response · Authors · 2023-11-22
>
> > (W3) ... how the connection model expressiveness to its adjacency function could drive the choice of such function... practical implication...
>
> A: Driving the choice of adjacency function is probably the most valuable consequence of this work. There are various adjacency function designs exist in the literature (See Appendix D). In general, most of the positional/structrual encoding and rewiring procedures, which brought huge improvements empirically, are an instantiation of adjacency functions. Previously, those are regarded as different directions of research. In this work, a general framework is proposed to setup a unified view on thoes encodings/procedure which brings extra expressiveness beyond 1-WL.

---

> ### Author Response · Authors · 2023-11-22
>
> > (Q1) ... How could this reconcile with the reported elevated efficacy of GTs ... any other potential empirical factors ...
>
> A: The reported elevated efficacy of GTs has also been questioned by [1] and [2]. Those papers raised two observations: 1. the globality of self-attention is more important than the feature-conditioning counterpart (query-key feature similarity). 2. The performance gain of GTs highly depends on positional encodings. Our theoretical framework and empirical result explain their findings while providing a deeper understanding of the elevated efficacy of GTs.
> Here we summarize our empirical findings: removing QK attention would not affect the performance of GTs but changing adjacent functions affect the performance a lot. In detail -- Table. 3, concise with theoretical finding, shows GPNN-PE (a GNN with non-feature-conditioning global filters based on a properly designed adjacency function) outperforms most previous GTs with global self-attention and only slightly underperform the SOTA GTs (GRIT) with the same adjacency function.
>
> **Thank you once more for your valuable and detailed feedback, which has significantly enhanced the work. The authors are committed to revising their paper, incorporating the recommended plots, and diligently addressing the highlighted issues and shortcomings. Upon your assessment that we have adequately resolved the identified concerns, we would appreciate your consideration of a re-evaluation of our work. Many thanks.**
>
>
> [1] Cai, C., Hy, T.S., Yu, R. &amp; Wang, Y.. (2023). On the Connection Between MPNN and Graph Transformer. Proceedings of the 40th International Conference on Machine Learning, 202:3408-3430.
>
> [2] Tönshoff, J., Ritzert, M., Rosenbluth, E., & Grohe, M. (2023). Where did the gap go? reassessing the long-range graph benchmark. arXiv preprint arXiv:2309.00367.

---

### Official Review · Reviewer_5UFG · 2023-11-10

**Soundness:** 3 good
**Presentation:** 3 good
**Contribution:** 2 fair
**Rating:** 3
**Confidence:** 4

**Summary:**

The paper proposed a general definition of generalized propagation graphs, which covers various graph neural networks based on different constructions of the adjacency function (see 3.2). The paper explored the expressiveness of GPNN, and an upper bound was proved. Plus, the paper defined CURC (continuous unified Ricci curvature) and utilized it to explore expressive power. At the end, the paper designed experiments to empirically prove their theory. By dropping the irrelevant part,  it shows graph network retaining the performance as original model, which in turn serves as an example that the graph network is primarily dependent on the adjacent function

**Strengths:**

1. Understanding a neutral network with strict mathematical theory is challenging. The paper proposed a general framework and some intuitive definitions to provide a general theoretical exploration of the graph neural network.

2. The paper designed experiments to validate its theoretical observation.

3. The paper provides proof for propositions and theorems it claims.

**Weaknesses:**

1. Since it is a theoretical-style paper, the paper needs to improve its notation and clarification, and it's better to treat these parts like math.
For the definition of adjacency function f, In section 3.2, 'it is a mapping from A to... ', so it is a mapping from R^{n^2} to R^{n^2 \times d}. But when referring to Appendix A1, it mentions some function that maps over some tensor power space from R^{n^p \times d} to R^{n^k \times d}. It seems that the adjacency function is a function on input with features, i.e., R^{n^2 \times d}. I am confused about which one should be.

2. The paper would be improved if it talked about why we chose Ricci curvature (generalized or not) as a measurement to explore the graph. In geometry, Ricci curvature is a degrade of Riemann curvature, while it is enough to characterize two-dimensional manifolds.

3. (Section 4.2, definition 4.2, d^{\epsilon} and (8)) The definition of CURC is questionable, which lies in the so-call asymmetric metric function. As a curvature to measure the curvedness of a manifold (in our case, a graph), its definition should be based on distance (or, say, metric) satisfying positivity, symmetry and triangle inequality. But CURC is based on an asymmetric metric function (which is not a metric), which means the metric could be different from u to v or v to u, which in turn means that the curvature from u to v could be more curved or less curved than the reverse. But CURC is defined as a scale, which contradicts the above.

4. The paper would be improved if it elaborated more on the implications of CURC, KR duality in section 4.3. It is hard to follow the terminology and abstract indications.

5. The definition of a generalized propagation network is more of a rewrite of a graph network with an adjacency function that incorporates propagation and feature mapping (there is still some ambiguity, as I mentioned in the first point). It is more on the conveniences to bring out the notion of curvature.

**Questions:**

1. (7) and lemma B7 in appendix: what does it mean m(x) divided by m(y)? as the definition of m just above, they are vectors.
2. CURC measuring GPNN (the paragraph above section 4.3). 'Furthermore, the incorporation of the Perron measure as .....'. How they relates to bottlenecking within weighted-directed graphs? any section it refers to?

---

> ### Author Response · Authors · 2023-11-22
>
> The authors would like to thank the reviewers for their constructive comments. Although this is a theoretical-style of work, they would like to emphasize that the theory serves as a tool and their main target is to provide such a tool for general research in this area. The authors will revise their paper to clarify the presentation accordingly. For the pointed out weakness and question, they would like to address as follows:

---

> > ### Author Response · Authors · 2023-11-22
> >
> > > (W3) ...The definition of CURC is questionable, which lies in the so-called asymmetric metric function ... CURC is based on an asymmetric metric function (which is not a metric) ... But CURC is defined as a scale, which contradicts the above.
> >
> > A: Thanks for the observations regarding the definition of CURC and its underlying asymmetric metric function in our paper. Your point about traditional curvature measures in geometry relying on symmetric metrics is well-founded. We would like to emphasize that CURC is a **scalar valued function** defined on **node-pairs (source and target)** as stated in definition 4.3, which does not contradict the asymmetric nature of directed distance on weighted-digraphs.
> >
> > The asymmetric nature of the "distance" used in CURC is a **deliberate and crucial aspect** of our approach, particularly when dealing with propagation graphs in Graph Transformers, which are naturally asymmetric. The asymmetry in the metric function allows CURC to capture the flow of information carried by the **directed and weighted propagation graphs**. This is essential in accurately measuring the propagation dynamics within Graph Transformers. Even though it is proper with symmetric input graphs, the traditional symmetric metrics would not adequately measure the propagation graphs for a better understanding of the behaviors of graph transformers.

---

> ### Author Response · Authors · 2023-11-22
>
> > (W1) Since it is a theoretical-style paper, ... improve notation and clarification... For the definition of adjacency function f, In section 3.2, 'it is a mapping from ... R^{n^2} to R^{n^2 \times d}. But when referring to Appendix A1, it mentions some function that maps ... from R^{n^p \times d} to R^{n^k \times d}.... I am confused about which one should be.
>
> A:
> **shared notion $f$** Appendix A1. is a general recap of permutation equivariant functions. The $f$ function in Appendix A1 is not the adjacency function. However, they share the same notion $f$ due to the adjacency function being permutation equivariant.
>
> **why not restricting Appendix A1 the same as the Adjacency function?** We choose to introduce permutation equivariance by defining its most general form, which is between two "arbitrary order" tensor power spaces. This is because a strong adjacency function could be realized/implemented by stacking mappings between such higher-order tensor power space. In fact, such implementation is proved to be crucial and unavoidable as showed in Equivaraint GNN [1] (Haggai) [Theorem 1] that a stronger expressiveness (k-FWL) needs at least R^(n^k) “intermediate variable” during the computation.  Since we are targeting to provide a general framework and considering the additional content of CURC, such an indication remains implicit due to the content limit. We thank the reviewers’ comments and include an extra section in the Appendix to discuss the designing space of the permutation equivariant adjacency function.

---

> ### Author Response · Authors · 2023-11-22
>
> > (W2) The paper would be improved if it talked about why we chose Ricci curvature (generalized or not) as a measurement to explore the graph. In geometry, Ricci curvature is a degrade of Riemann curvature, while it is enough to characterize two-dimensional manifolds.
>
> A: This comment is regarding towards **our choice of Ricci curvature**, in the context of exploring graphs in our paper. Your observation about the role of Ricci curvature in geometry is indeed accurate. Ricci curvature, derived from the more general Riemann curvature tensor, is crucial in the characterization of two-dimensional manifolds. However, our paper's focus extends beyond the traditional geometric interpretation of Ricci curvature to its application in graph analysis, particularly within the framework of Graph Transformers.
>
> In our study, we build upon existing research that employs Ollivier-Ricci curvature for analyzing graphs [2]. The choice of Ollivier-Ricci curvature is motivated by its ability to capture the geometric properties of a graph in a way that is particularly relevant to our analysis of Graph Transformers. Specifically, CURC extends the application of Ollivier-Ricci curvature to weighted directed graphs (digraphs), which are representative of the propagation graphs defined in our GPNN framework, unifying various GNNs (e.g., MPNNs, graph rewiring, graph transformers).
> The extension to weighted digraphs is not trivial because it allows for a more nuanced understanding of how GNNs behave via propagation graphs rather than on the property of input graphs,
> where the directionality and weights of edges play a critical role. This is particularly pertinent in the context of Graph Transformers, where the learned propagation graphs can differ from the input graphs remarkably.
>
> We believe this extension not only enriches the theoretical understanding of graph curvature in the realm of machine learning and graph neural networks but also provides practical benefits in terms of enhanced model interpretability and performance.

---

> ### Author Response · Authors · 2023-11-22
>
> > (W4) The paper would be improved if it elaborated more on the implications of CURC, KR duality in section 4.3. It is hard to follow the terminology and abstract indications.
>
>
> A: Thank you for pointing out the areas in section 4.3 where the discussion on CURC and KR duality could be more comprehensive. We acknowledge that the current terminology and abstract indications may be challenging to follow. In response, we are committed to refining the terminology and clarifying the indications for reader-friendliness. Additionally, we encourage readers to refer to Appendix B, where these concepts are explained in detail. This should provide a more thorough understanding of the topic.
>
> > (W5) ... generalized propagation network is a rewrite of a graph network with an adjacency function ... It is more on the conveniences to bring out the notion of curvature.
>
> A: We understand your concern that the current definition might appear as a mere reiteration of a graph network with an enhanced adjacency function. However, our intention was to highlight the GPNN framework as a significant contribution in its own right, serving as a unifying model in the realm of graph deep learning. With the GPNN framework, we can analyze the expressiveness of various GNNs from a unified approach and understand the dominant components in designing more expressive GNNs.

---

> ### Author Response · Authors · 2023-11-22
>
> > (Q1)  (7) and lemma B7 in appendix: what does it mean m(x) divided by m(y)? as the definition of m just above, they are vectors.
>
> A: In response to your query about the expression $m(x)$ divided by $m(y)$ in (7) and Lemma B7 of the appendix, it's important to clarify the nature of the mapping $m$. As outlined in Definition 4, $m$ is indeed a mapping from the vertex set $V$ to the interval $(0,1]$. This mapping places $m$ in the space $R^{(0, 1] \times |V|}$, considering the discrete nature of the vertex set.
>
> The normalized left eigenvector, which we refer to in our paper, resides within this space. Therefore, when we mention $ m(x) $, we are specifically referring to the x-th value of this eigenvector, denoted as $ (\frac{v\_{pf}}{||v\_{pf}||})\_x $. This notation represents the x-th component of the eigenvector $ v_{pf} $ normalized by its norm $ ||v_{pf}|| $.
>
> When discussing $ m(x) $ divided by $ m(y) $, we are comparing these specific components of the normalized eigenvector, which correspond to different vertices in the graph respectively. This comparison is crucial for the analysis presented in section 7 and Lemma B7, as it relates to the behavior of the network at different vertices.

---

> ### Author Response · Authors · 2023-11-22
>
> > (Q2) CURC measuring GPNN (the paragraph above section 4.3). 'Furthermore, the incorporation of the Perron measure as .....'. How they relates to bottlenecking within weighted-directed graphs? any section it refers to?
>
> A: The notion of a "bottleneck" in undirected graphs has been extensively studied and is known to be a critical factor affecting the performance of graph models [2]. In such models, bottlenecking refers to a situation where a small number of edges or vertices significantly restrict the flow or connectivity within the network. The authors' proposal in [2] to use Ollivier-Ricci (OR) curvature on input graphs as a means to quantify this phenomenon has been a significant advancement in understanding graph structures.
> When we extend the concept of bottlenecking to weighted-directed graphs, the CURC (Curvature Under Ricci Curvature) becomes a valuable tool.
> This is driven by the fact that different GNNs (e.g., MPNNs, graph rewiring, graph transformers) behave differently given the same input graphs.
> Alternatively, we propose to analyze the propagation graphs of various GNNs under the GPNN framework, which are naturally weighted-directed graphs and not compatible to OR curvature.
> In this context, CURC's extension to OR curvature allows for a nuanced understanding of bottleneck phenomena in the propagation graphs. This generalization is crucial as it adapts the concept of bottlenecks, traditionally applied to undirected graphs, to the realm of weighted-directed graphs, thereby providing deeper insights into the bottleneck of GNNs. We will include a more throughout discussion on this topic in the appendix
>
> **The authors hope this clarification addresses the reviewer's query and highlights the novelty and significance of their approach in integrating geometric concepts with advanced graph analytical methods.**
>
>
>
> [1] Maron, H., Ben-Hamu, H., Serviansky, H., & Lipman, Y. (2019). Provably powerful graph networks. Advances in neural information processing systems, 32.
>
> [2] Topping, J., Di Giovanni, F., Chamberlain, B. P., Dong, X., & Bronstein, M. M. (2021, October). Understanding over-squashing and bottlenecks on graphs via curvature. In International Conference on Learning Representations.

---

### Meta-Review · Area_Chair_V5zQ · 2023-12-05

**Metareview:**

This paper introduces a unified framework for various variants of graph neural networks (from MPNNs, graph rewiring, and SubgraphGNNs to GTs), called generalized propagation graphs, by modeling them with the adjacency function and the entry-wise function. Based on this framework, they analyze the graph expressiveness by an upper bound. And then they analyze the bottleneck problem with an extension of the Ollivier-Ricc curvature applicable to any strongly-connected weighted-directed graph, called CURE.  They study the theoretical properties of CURC and leverage this in analyzing the shifts in curvature distribution for propagation graphs after training, for a number of graph learning models that can be cast as GPNNs.

Reviewers generally agree that the unification of different GNNs using the same framework is an important topic, and there are some theoretical analyses derived from the framework. On the other side, a common concern rises that the paper’s readability can be significantly improved, and the significance of the derived results is limited. Indeed, the paper presents an ambitious goal to unify different methods, but rewriting them as the adjacency function and the entry-wise function does not yield significantly novel results beyond existing literature. Also, the two parts, GPNN and CURE, are only loosely connected and it is not fully clear why the new CURC extension has to be involved in the GPNN framework (more motivation and connection needed). Generally, there are concerns shared by the reviewers that this paper lacks a clear and significant contribution towards a better understanding of graph neural networks. The authors have responded to the concerns of the reviewers, but since no one replied (unfortunately), I suppose that the concerns still remain.

Summarizing these opinions, I would suggest rejecting this paper in its current form, since it fails to meet the ICLR bar. The authors are strongly encouraged to rewrite the paper — to be more specific on the theoretical contributions, and to be clearer on the storyline that links different parts of the paper — and submit it to a future venue.

**Justification For Why Not Higher Score:**

There are shared concerns among reviewers that this paper is generally hard to follow and has limited significant results.

**Justification For Why Not Lower Score:**

N/A

---

### Decision · Program_Chairs · 2024-01-16

Reject